# A regulatory microRNA network controls endothelial cell phenotypic switch during sprouting angiogenesis

**Stefania Rosano[1,2], Davide Corà[3,4], Sushant Parab[1,2], Serena Zaffuto[1,2], Claudio Isella[1,2], Roberta Porporato[2], Roxana Maria Hoza[1,2], Raffaele A Calogero[5], Chiara Riganti[1], Federico Bussolino[1,2†]\*, Alessio Noghero[1,2†‡]\***

[1]Department of Oncology, University of Turin, Candiolo, Italy; [2]Candiolo Cancer Institute FPO-IRCCS, Candiolo, Italy; [3]Department of Translational Medicine, Piemonte Orientale University, Novara, Italy; [4]Center for Translational Research on Autoimmune and Allergic Diseases - CAAD, Novara, Italy; [5]Molecular Biotechnology Center, Department of Biotechnology and Health Sciences, University of Turin, Turin, Italy

**\*For correspondence:**
federico.bussolino@unito.it (FB);
anoghero@lrri.org (AN)

[†]These authors contributed equally to this work

**Present address:** [‡]Lovelace Biomedical Research Institute, Albuquerque, United States

**Competing interests:** The authors declare that no competing interests exist.

**Abstract** Angiogenesis requires the temporal coordination of the proliferation and the migration of endothelial cells. Here, we investigated the regulatory role of microRNAs (miRNAs) in harmonizing angiogenesis processes in a three-dimensional in vitro model. We described a microRNA network which contributes to the observed down- and upregulation of proliferative and migratory genes, respectively. Global analysis of miRNA–target gene interactions identified two sub-network modules, the first organized in upregulated miRNAs connected with downregulated target genes and the second with opposite features. miR-424–5p and miR-29a-3p were selected for the network validation. Gain- and loss-of-function approaches targeting these microRNAs impaired angiogenesis, suggesting that these modules are instrumental to the temporal coordination of endothelial migration and proliferation. Interestingly, miR-29a-3p and its targets belong to a selective biomarker that is able to identify colorectal cancer patients who are responding to anti-angiogenic treatments. Our results provide a view of higher-order interactions in angiogenesis that has potential to provide diagnostic and therapeutic insights.

## Introduction

The expansion of a vascular network through the sprouting angiogenesis (SA) process requires a coordinated control of many cellular functions, including the activation of quiescent endothelial cells (ECs), cell protrusion, basal lamina and extracellular matrix degradation, cell migration and proliferation, deposition of new basement membrane, cell junctions and cell polarity alteration (*Carmeliet and Jain, 2011*). In response to an angiogenic stimulus, activated ECs acquire distinct specialized phenotypes to accomplish these different tasks (*Jakobsson et al., 2010*), ultimately leading to the formation of a new functional vascular network. In particular, during SA, ECs dynamically switch from a tip phenotype, which guides the network expansion, to a stalk cell state that is characterized by active proliferation.

Vascular Endothelial Growth Factor-A (VEGF-A) is the most prominent cytokine that activates a genetic program that sustains this morphogenetic process (*Simons et al., 2016*). Transcriptomic analysis of tip and stalk cells isolated from the retinal vascular plexus showed differential expression of molecular determinants, represented by genes that are involved in extracellular matrix remodeling for tip cells and that encode components of the Notch signaling pathway for stalk cells (*del Toro*

*et al., 2010*; *Strasser et al., 2010*). It also has been proposed that the EC fate decision towards the tip or stalk cell phenotype is affected by VEGF-A gradient concentration in the surrounding extracellular matrix and by VEGF receptors expression levels (*Gerhardt et al., 2003*). Nevertheless, the gene expression regulatory events that sustain the initial transition from quiescent to activated ECs in response to an angiogenic stimulus have not yet been fully elucidated. We hypothesized that such phenotypic transition would require profound transcriptomic alterations that are brought about through the coordination of multiple regulation layers, including microRNA-mediated post-transcriptional regulation.

MicroRNAs (miRNAs) are a class of small non-coding RNAs that extensively modulate gene expression at the post-transcriptional level by targeting the mRNAs of protein-coding genes, directing their repression through mRNA degradation or (to a lesser extent) inhibition of protein translation (*Guo et al., 2010*). Interestingly, one of the first functions attributed to miRNAs was the control of developmental processes in which a morphogen gradient directs cell fate by activating specific genetic programs (*Inui et al., 2012*; *Ivey and Srivastava, 2015*). In this scenario, miRNAs are an integral part of the regulatory network that allows transitions between different cell states (*Hornstein and Shomron, 2006*). A role for miRNAs in ECs biology and vascular development has been established in vitro and in vivo by inhibition of two endonucleases that are required for mature miRNAs generation, namely Dicer and Drosha (*Ha and Kim, 2014*). The consequent reduction in the level of miRNAs resulted in alteration of several key properties of ECs, including their sprouting ability (*Kuehbacher et al., 2007*), and impaired postnatal angiogenesis (*Suárez et al., 2008*).

The activity of specific miRNAs in connection with the VEGF signaling pathway components has been studied more extensively (*Dang et al., 2013*). Several studies have demonstrated that miR-15a and miR-16, among other miRNAs, repress VEGF expression (*Chamorro-Jorganes et al., 2011*; *Yin et al., 2012*), whereas miR-126 promotes VEGF signaling by targeting the downstream effector *PIK3R2* (*Fish et al., 2008*). Furthermore, miR-27b and miR-221 are required for tip cell specification (*Biyashev et al., 2012*; *Nicoli et al., 2012*). Recently, RNA-sequencing (RNAseq) technology allowed the generation of a complete annotation of the miRNAs that are expressed by two-dimensional cultured human ECs in normal (*Kuosmanen et al., 2017*) or hypoxic (*Voellenkle et al., 2012*) conditions. Yet, the extent to which miRNAs could affect ECs phenotypic specification during SA has not been fully captured to date. Using RNAseq technology and network analysis, we exploited a three-dimensional model of SA that specifically describes the lateral inhibition-driven tip cell selection (*Heiss et al., 2015*; *Nowak-Sliwinska et al., 2018*), which is considered to be the first step in capillary nascence (*Eilken and Adams, 2010*). The information obtained was used to generate a co-expression network encompassing the post-transcriptionally regulated interactions between modulated miRNAs and their predicted protein-coding gene targets. Here, we show that in the initial step of SA, miRNAs act cooperatively to give robustness to the specification of the tip cell phenotype by reducing the expression of genes that are associated with cell-cycle progression and of members of the mitogen-associated protein kinase (MAPK) cascade that sustains VEGF-A-mediated cell proliferation, while de-repressing genes that are involved in cell migration and extracellular matrix remodeling.

## Results

### VEGF-A induces the tip phenotype of endothelial cells in a 3D model of sprouting angiogenesis

To study the activation of quiescent endothelial cells induced by an angiogenic stimulus, and the impact that miRNAs may exert on this process, we exploited a three-dimensional (3D) model that mimics the initial phase of SA in vitro (*Heiss et al., 2015*; *Nowak-Sliwinska et al., 2018*). ECs were induced to form 3D spheroids that are characterized by mature EC–EC junctions that are responsible for quiescent proliferation state (*Weidemann et al., 2013*). Spheroids were then embedded in a 3D collagen matrix and exposed to VEGF-A to trigger SA and to stimulate the formation of capillary-like structures (*Figure 1A*). Spheroids that had been exposed to VEGF-A for 18 hr (SPHV) and control spheroids (SPHC) were then processed for long- and small-RNAseq. Expression analysis of protein-coding genes identified 3071 differentially expressed (DE) genes in SPHV compared to SPHC (*Figure 1—figure supplement 1A*), revealing dramatic changes in the global transcriptomic profile

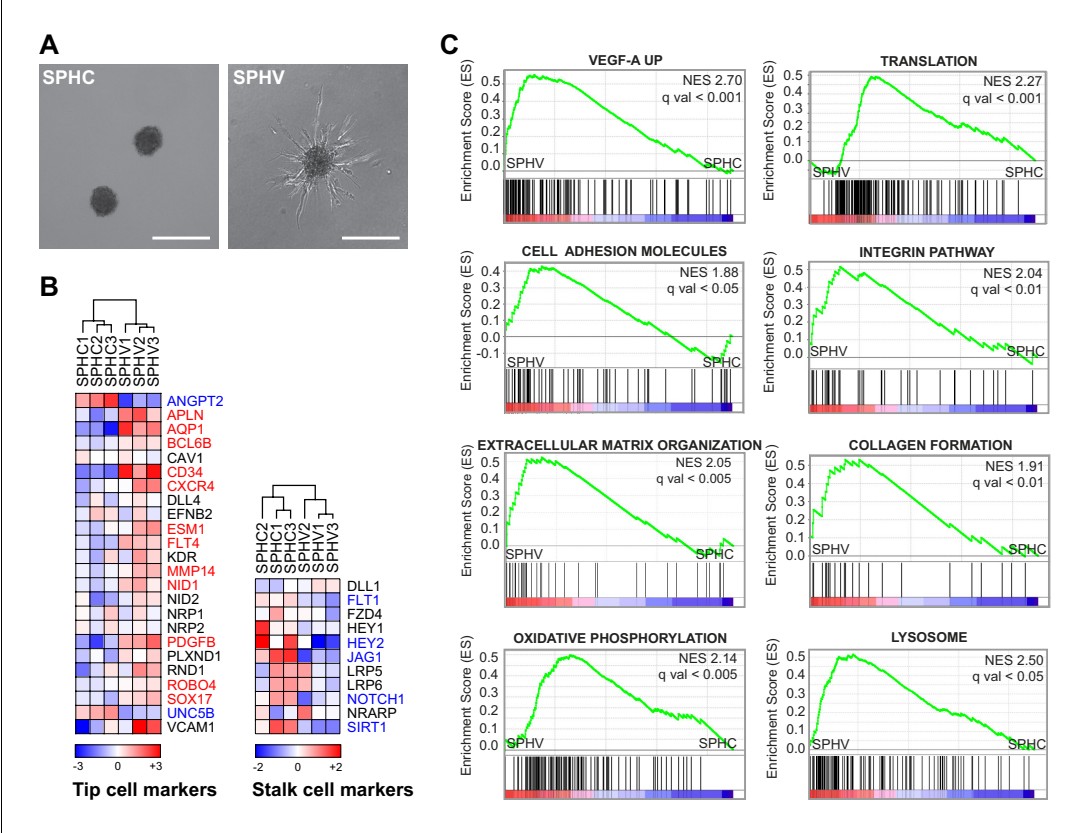

**Figure 1.** VEGF-A induces the tip phenotype of endothelial cells in a 3D model of sprouting angiogenesis. (**A**) EC spheroids were embedded in a collagen matrix and stimulated with VEGF-A to induce endothelial cell sprouting. RNA-sequencing analysis was performed on control spheroids (SPHC) not stimulated with VEGF-A, and on spheroids after 18 hr from stimulus (SPHV). Scale bars, 200 μm. (**B**) Heatmaps showing the expression of tip and stalk cell markers in SPHC and SPHV. Color bars indicate log$_2$(fold change) (log$_2$FC) in the comparison SPCV versus SPHC. (**C**) Positively enriched gene sets identified by the GSEA study in the genesets collection for canonical pathways. False discovery rate (FDR) was accepted when q value <0.05. The online version of this article includes the following figure supplement(s) for figure 1:

**Figure supplement 1.** Expression profile of protein-coding genes annotated from the RNA-sequencing data.
**Figure supplement 2.** Organization of tip and stalk cells and the effect of Notch pathway manipulation in EC spheroids.

during the early phase of sprouting. Upregulation of VEGFR2, the major VEGF signal transducer, was validated by real-time PCR and Western blot analysis (*Figure 1—figure supplement 1B–D*). By looking at the expression of known tip and stalk cell markers (*De Bock et al., 2013*; *del Toro et al., 2010*; *Strasser et al., 2010*), we observed a general upregulation of tip marker genes and a downregulation of stalk marker genes in the comparison between SPHV and SPHC (*Figure 1B*). To visualize the organization of tip and stalk cells in this model, we analyzed the tip cell markers DLL4 and CXCR4 by immunofluorescence, or we transduced ECs with a fluorescent reporter of Notch activity to visualize stalk cells (*Figure 1—figure supplement 2A*). Although the EC sprouts strongly expressed DLL4 and CXCR4, cells in which the Notch reporter was activated were present only occasionally, so ECs largely acquired a tip-cell phenotype. Furthermore, manipulation of the Notch pathway by the γ-secretase inhibitor DAPT or by exogenous DLL4 resulted in increased or reduced sprouting, respectively (*Figure 1—figure supplement 2B*). Altogether, these observations indicate that this sprouting model can be used, at this specific developmental time point, to represent the early phase of VEGF-A response that involves activation of quiescent ECs and differentiation to tip cells.

To identify key biological pathways or that set of genes that is relevant to the sprouting process, the gene expression dataset was interrogated by taking advantage of Gene Set Enrichment Analysis (GSEA) (*Subramanian et al., 2005*). When comparing SPHV with SPHC, this analysis confirmed the upregulation of known VEGF-A responder genes ('VEGF-A UP'), and revealed a positive enrichment

of gene sets that are representative of several biological pathways and classes of molecules, including 'Translation', 'Cell adhesion molecules', 'Integrin pathway', 'Extracellular matrix organization', and 'Collagen formation'. Altogether, these alterations in gene expression mirror the transcriptional landscape that allows the migratory process. Other representative enriched pathways included 'Oxidative phosphorylation' and 'Lysosome' (*Figure 1C*).

## Early VEGF-A response does not involve cell proliferation

Activation of the VEGF pathway through VEGF receptors initiates a signaling cascade that promotes the proliferation of ECs, among other actions (*Simons et al., 2016*). Nonetheless, and in agreement with the non-proliferating tip cell phenotype, GSEA showed a strongly negative enrichment score (ES) for cell-cycle-controlling genes (*Figure 2A*). Indeed, when looking at the expression levels of genes belonging to the 'cell cycle' KEGG pathway (*Kanehisa et al., 2016*), expression analysis showed that 52 genes out of 113 were DE in SPHV compared to SPHC, of which 50 were downregulated (*Figure 2B*). Hypergeometric test confirmed a statistically significant over-representation of this subset of genes among all the DE genes ($P = 7.40 \times 10^{-5}$). Only cyclin D2 (CCND2) and cyclin E1 (CCNE1), two genes belonging to the cyclin family and participating in the $G_1$phase of the cell cycle, were upregulated. The finding that VEGF-A did not activate the proliferation of ECs and the accompanying DNA synthesis was also confirmed by the negative ES of gene sets that are representative of pyrimidine metabolism (*Figure 2C*) and, to a lesser extent, purine metabolism (*Figure 2D*). Indeed, we observed a general downregulation of genes involved in de novo synthesis of nucleotides (*Figure 2—figure supplement 1A*). Downregulations of the cyclin B1 gene and of IMPDH2, the rate-limiting enzyme in de novo guanine nucleotide biosynthesis, were validated by a real-time PCR assay (*Figure 2—figure supplement 1B,C*). We then measured the activities of the enzymes PPAT and CAD, which are considered to be indices of de novo synthesis of purine and pyrimidine nucleotides and of ATIC, which catalyzes the last step of purine synthesis, respectively. VEGF-A stimulation decreased the activity of these three enzymes in the 3D model (*Figure 2E*) indicating that, in this condition, de novo nucleotide synthesis is not required. On the contrary, their activity significantly increased in a control experiment performed with ECs cultured in 2D conditions (*Figure 2—figure supplement 1D*). Measurement of the percentage of cells that entered the S phase of the cell cycle confirmed that SPHC displayed a quiescent, non-proliferative status, and that VEGF-A was not able to trigger cell proliferation in spheroids significantly within an 18-hr time frame (*Figure 2F*). By contrast, VEGF-A activity on cell proliferation was well evident in cells cultured in standard 2D conditions, both in a proliferation assay and in a GSEA of the corresponding microarray data (*Figure 2—figure supplement 1E,F*). Furthermore, ECs that were mitotically inactivated by mitomycin C treatment maintained the ability to migrate and to form endothelial sprouts (*Figure 2G*). Taken together, these observations indicate that, in 3D conditions, early VEGF-A activity is primarily directed towards cell migration rather than towards cell proliferation.

## A miRNA-dependent regulatory network sustains acquisition of the tip cell phenotype

Analysis of the small-RNAs sequencing data allowed the annotation of 639 mature miRNAs that were expressed by ECs across the two experimental conditions (SPHC and SPHV), at levels above a fixed cutoff (raw counts = 5) (*Figure 3—source data 1*). To assess the impact of endogenous miRNAs on the phenotypic differentiation that occurs during SA, we applied a bioinformatics pipeline based on the co-expression analysis of miRNAs and protein-coding genes, the prediction of miRNA target genes, and the association of miRNAs with biological pathways, followed by network analysis (*Figure 3A*). We first performed pair-wise correlation analysis between miRNAs and the expression of protein-coding expression, and plotted the distribution of the correlator (*Figure 3B*). We then isolated the subset of pairs that contained evolutionarily conserved or non-conserved miRNA–target gene interactions, based on TargetScan predictions (*Agarwal et al., 2015*). The frequency distribution of the correlator for the conserved interactions subset showed a significant enrichment in negative correlations compared to that for the entire dataset (*Figure 3B*, blue line). This is indicative of the repressive activity exerted by miRNAs on their targets; that is, when a miRNA is upregulated, its protein-coding gene target is downregulated and vice-versa. By contrast, the

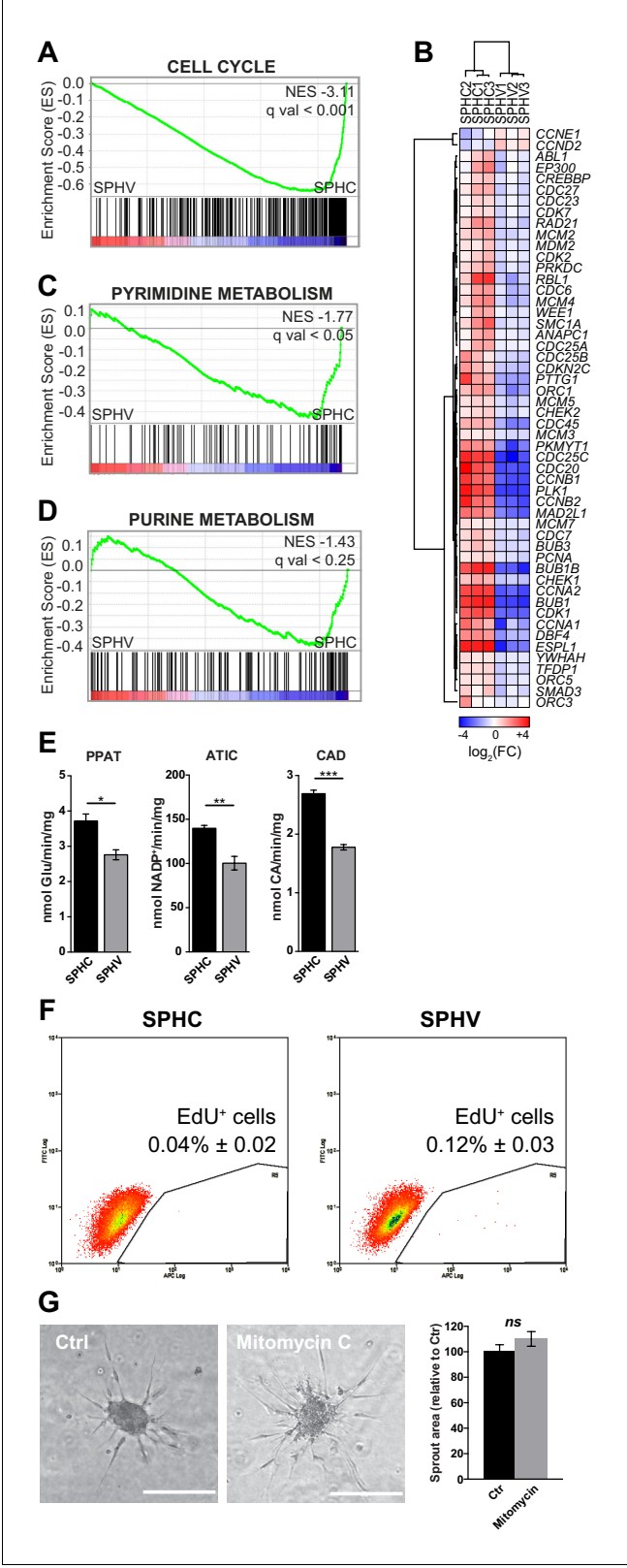

**Figure 2.** Early VEGF-A response does not involve cell proliferation. (**A**) GSEA plot showing negative association with the 'cell cycle' gene set in SPHV compared to SPHC. (**B**) Heatmap representing the expression of genes that are included in the KEGG pathway 'cell cycle' and are differentially expressed with a log$_2$(Fold Change)>0.5, FDR < 0.05 when comparing SPHV versus SPHC. Color bar indicates log$_2$(fold change). (**C, D**) GSEA plots showing,

*Figure 2 continued on next page*

*Figure 2 continued*

respectively, negative association with 'pyrimidine metabolism' and 'purine metabolism' gene sets in SPHV compared to SPHC. FDR was accepted when q value < 0.05. (E) Measure of the VEGF-A-induced enzymatic activity of three enzymes involved in de novo nucleotide synthesis in SPHC and SPHV. PPAT, phosphoribosyl pyrophosphate amidotransferase; ATIC, 5-aminoimidazole-4-carboxamide ribonucleotide formyltransferase/IMP cyclohydrolase; CAD, carbamoyl-phosphate synthetase 2/aspartate transcarbamylase/dihydroorotase. Data are represented as mean ± SEM of n = 3 experiments. (F) Cell proliferation rate analysis in SPHC and SPHV performed by evaluation of EdU incorporation into the newly synthetized DNA. Representative plots of n = 3 experiments. Data represent the mean percentage of proliferating cells ± SEM of n = 3 experiments. p<0.05 in the comparison between SPHC and SPHV. (G) Sprouting assay performed with ECs pre-treated with the cell replication blocker mitomycin C, and the corresponding quantification of sprout area. Data are represented as mean ± SEM from n = 3 experiments. ***, p<0.001; **, p<0.01; *, p<0.05; *ns*, not significant.

The online version of this article includes the following figure supplement(s) for figure 2:

**Figure supplement 1.** Expression profile of genes involved in nucleotide synthesis and cell proliferation assays.

---

non-conserved interactions subset did not show any significant alteration in the correlator frequency distribution (*Figure 3—figure supplement 1A*). Considering that target-site conservation is believed to be an indication of functional repression for mammalian miRNAs (*Friedman et al., 2009*), the integration of the SA gene expression dataset with information about evolutionarily conserved miRNA–target gene interactions provides information on the global activity of miRNAs taking place during the early sprouting phase. Therefore, this strategy permits the identification of target genes and biological processes that are more likely to be affected by miRNA-mediated post-transcriptional regulation.

GSEA-based analysis of correlation between miRNA expression and groups of genes participating in the regulation of specific biological pathways, represented as gene sets, was also used to support our findings (*Figure 3C*). This analysis identified five clusters of genesets representing distinct biological functions that are modulated during SA, together with their associated miRNAs. Gene ontology-based enrichment analysis performed on the genes contained in each cluster showed the following classes: 1) extracellular matrix remodeling and cell migration, 2) protein translation and metabolic processes, 3) intracellular signaling pathways, 4) RNA and protein processing, and, 5) cell cycle, DNA repair and MAPK cascade (*Figure 3D*). Clusters 1 and 2 contained genes that are mostly upregulated in SPHV compared to SPHC, whereas clusters 3, 4 and 5 contained genes that are mostly downregulated. To generate the global post-transcriptional regulatory network that sustains SA, the results from the above analyses were integrated. The resulting miRNA-centered co-expression network is composed of 149 miRNAs and 717 protein-coding genes connected by 1713 edges (*Figure 4* and *Figure 4—source data 1*). This network is essentially characterized by two individual components: one representing upregulated miRNAs connected with downregulated target genes, and one representing downregulated miRNAs connected with upregulated target genes. Degree distribution for both miRNAs (out-degree) and protein-coding genes (in-degree) observed a power-law behavior (*Figure 3—figure supplement 1B,C*), indicating that this network is compatible with the scale-free hypothesis, a typical feature of biological networks (*Barabási and Pósfai, 2016*).

## The ERK gene module is under the control of a miRNA network in sprouting angiogenesis

VEGF-mediated EC proliferation and migration recognize the ERK pathway (*Wong and Jin, 2005*) and p38 MAP kinase activity (*Rousseau et al., 2000*), respectively, as master signaling modules (*Simons et al., 2016*). The data shown so far demonstrate that genes controlling cell proliferation were strongly downregulated (*Figure 2*), as were the expression levels of miRNAs correlated with genes associated with cell division and the MAPK cascade (*Figure 3D*). On these bases, we asked whether miRNAs have a role in suppressing the proliferative signal downstream VEGF-A through ERK pathway inhibition, while supporting cell migration in a 3D context. To this purpose, we interrogated the previously defined global post-transcriptional network (*Figure 4—source data 1*), and extracted the nodes corresponding to genes that encode known MAP kinases (*Craig et al., 2008*) together with their targeting miRNAs. Mapping of these interactions generated a network composed of one major component, which represents the inhibitory activity of miRNAs over several

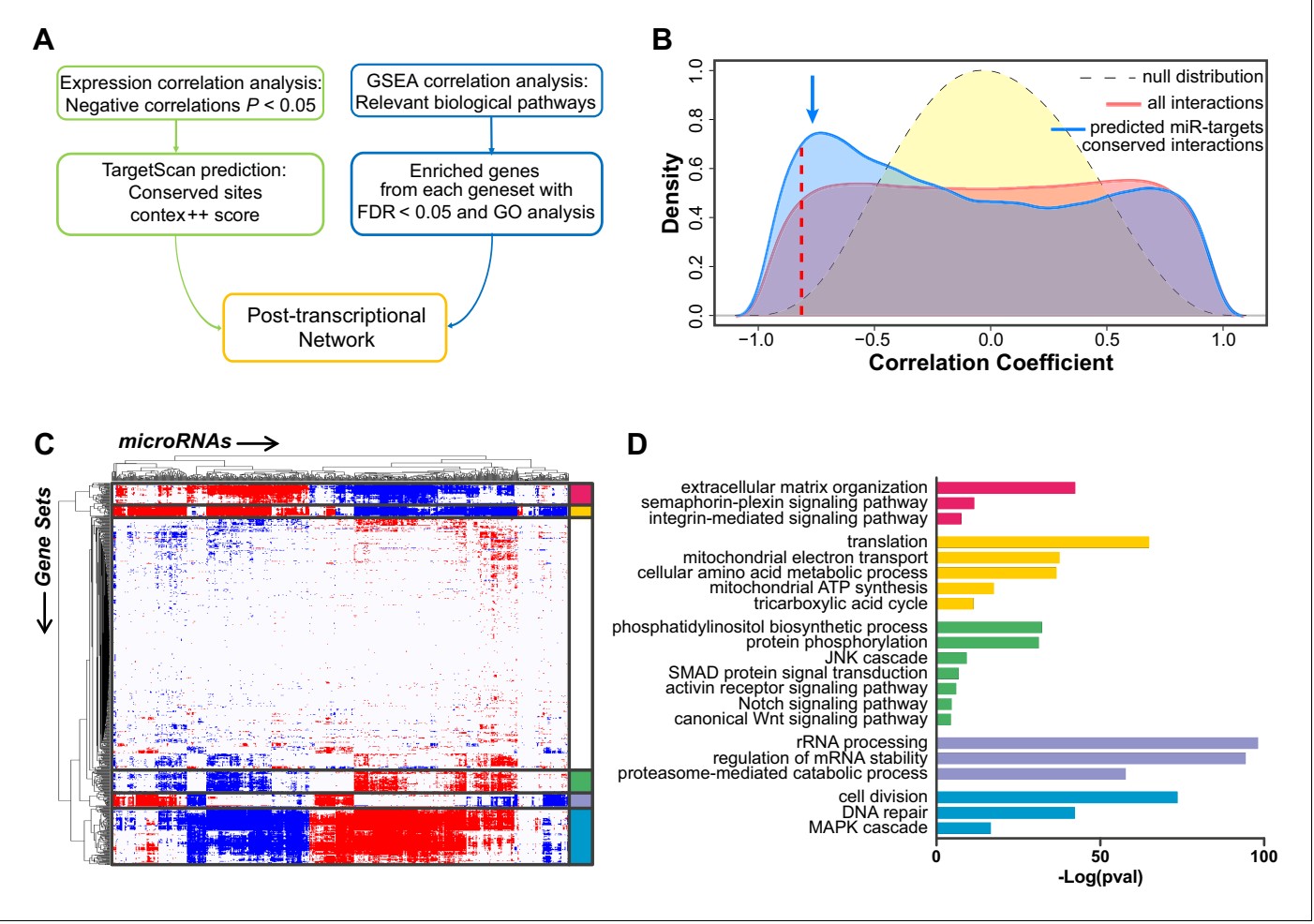

**Figure 3.** Construction of the post-transcriptional regulatory network that sustains acquisition of the tip cell phenotype. (**A**) Schematic view of the computational approach used to generate the post-transcriptional regulatory network. (**B**) Distribution of the correlator (Pearson coefficient) of the pair-wise analysis that considered the expression profile of all miRNAs and of protein-coding genes expressed in SPCH and SPHV. The gray dashed curve represents the null hypothesis distribution (uncorrelated miRNA–protein coding gene pairs) obtained by sample randomization with 1000 permutations; the red curve represents all pair-wise miRNA–protein-coding gene interactions; the blue curve represents the subset of miRNA–protein-coding gene pairs containing an evolutionarily conserved interaction as predicted by the TargetScan database. The blue arrow points to the enrichment in negative correlations. The red dashed line indicates the statistical significance threshold chosen for the correlator ($P < 0.05$). (**C**) Association matrix of the expression profiles of miRNAs (columns) with functional gene sets (rows) representing canonical pathways. Significant associations (FDR < 0.05) are shown in red (positive) or blue (negative). White, not significant. Biologically relevant clusters are highlighted with different colors. (**D**) For each cluster in (**C**), genes contributing to enrichment in the correlated gene sets were analyzed by functional annotation. Graph shows –Log(*P* value) of representative gene ontology terms.

The online version of this article includes the following source data and figure supplement(s) for figure 3:

**Source data 1.** miRNAs annotation.

**Figure supplement 1.** Non-conserved miRNA-target gene interactions analysis and network degree analysis.

MAP kinases (*Figure 5A*). These MAP kinases showed reduced expression in SPHV compared to SPHC. The *TAOK1* gene, whose protein product is a kinase that activates stress response (*Raman et al., 2007*), and *MAPK1*, whose protein product is ERK2, were the most targeted genes in this sub-network, being targeted by 11 and 7 different miRNAs, respectively. Notably, all components constituting the ERK module, namely C-Raf (*RAF1*), MEK (*MAP2K1*) and ERK2 (*MAPK1*), were represented. Among the four members composing the p38MAPK family (p38α, -β, -γ and -δ), only p38α (*MAPK14*) was present, but this gene was not connected with the major component of the network because its expression is increased in SPHV compared to SPHC. Downregulation of *RAF1*, *MAP2K1*, *MAPK1* and *TAOK1*, and upregulation of *MAPK14* in SPHV compared to SPHC, was

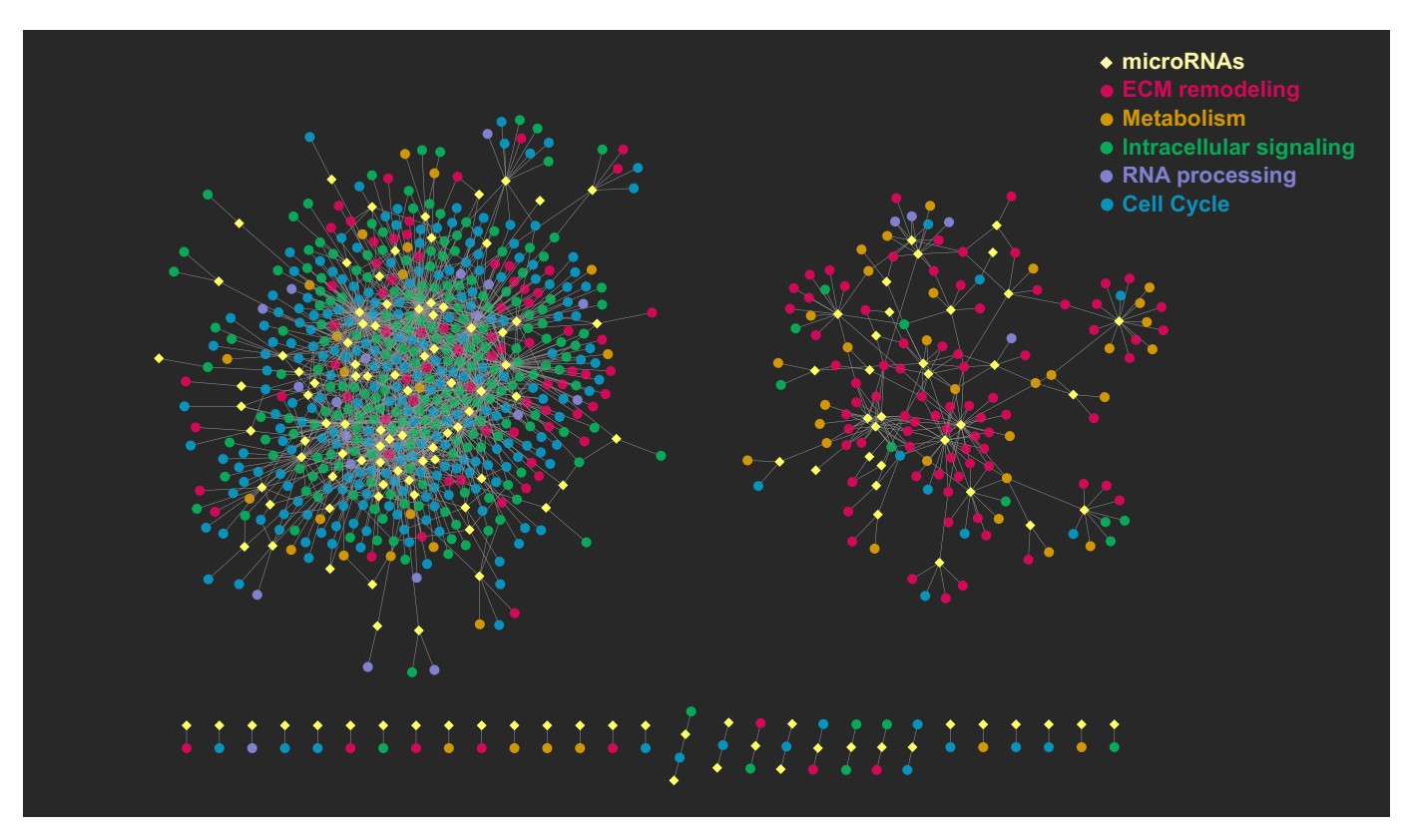

**Figure 4.** Graphical representation of the post-transcriptional network. The network consists of two major, independent components. The component on the left includes the interactions between upregulated miRNAs and their respective downregulated target genes; the component on the right includes the interactions between downregulated miRNAs and their respective upregulated target genes. Yellow diamonds represent miRNAs, circles represent their target genes, colored on the basis of the gene ontology cluster to which they belong. Edges are the miRNA–target gene pairs with a predicted direct interaction and a significant negative correlation.

The online version of this article includes the following source data for figure 4:

**Source data 1.** Co-expression network edges.

validated by real-time PCR (*Figure 5B*). Furthermore, ERK activity was measured in SPHC and SPHV by Meso Scale Technology (MSD) and expressed as P-ERK/total ERK ratio in order to take variations in total ERK expression into account. *Figure 5C* shows that ERK activity decreases after VEGF treatment. We also analyzed the effect in the sprouting model of SHC 772984 or SB 202190 compounds, which inhibit ERK1/2 and p38MAPK, respectively. Effective inhibitor concentrations were assessed by evaluating the inhibitory activity on VEGF-A-induced enzyme phosphorylation (*Figure 5—figure supplement 1*). As shown in *Figure 5D*, treatment with the ERK inhibitor SCH 772984 did not have any effect on sprouting, confirming that ERK activity is not required, and in agreement with the observed low proliferation rate (*Figure 2*). By contrast, treatment with p38MAPK inhibitor SB 202190 completely abrogated the ability of ECs to migrate and to form sprouts.

### *DICER* knock-down rescues the proliferative effect of VEGF-A

To demonstrate a regulatory role exerted by miRNAs on their predicted target genes and, more generally, to confirm the involvement of miRNAs in the control of the sprouting process, we removed mature miRNAs from the system by inactivating DICER, one of the endonucleases responsible for the generation of mature miRNAs from their precursors (*Ha and Kim, 2014*). This approach, which investigated the effects of miRNAs on the global post-transcriptional network, was preferred over the inhibition of individual miRNAs. Cells in which *DICER* expression was knocked-down by a short-hairpin RNA (*DICER*[KD]) showed a reduction of sprouting to about 50% of that seen in cells

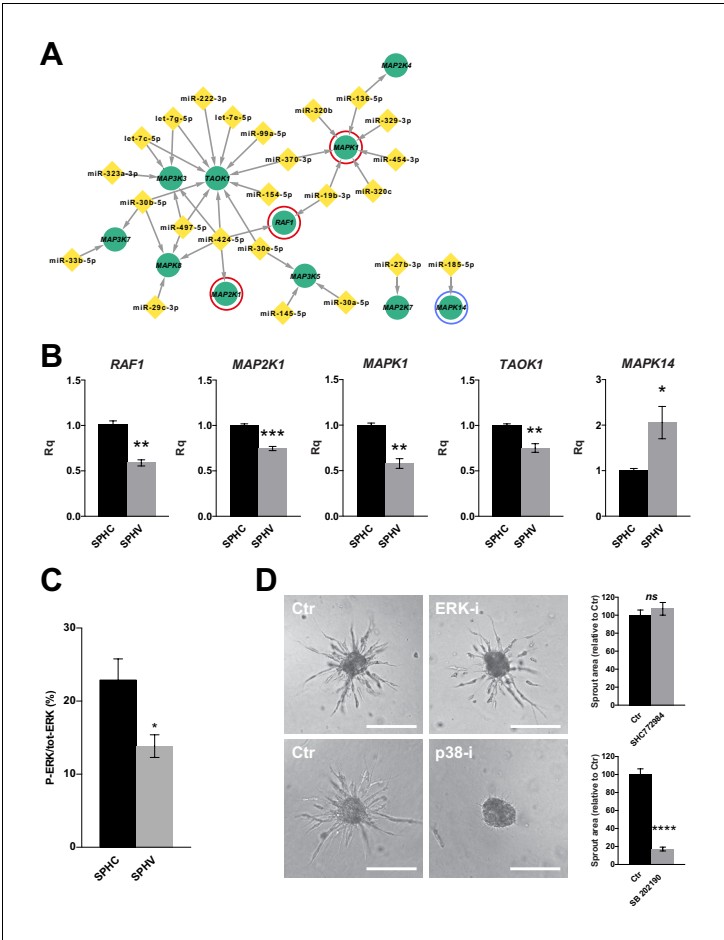

**Figure 5.** ERK activation is not required for sprouting and the ERK module is repressed by miRNAs. (**A**) Sub-network representing miRNAs targeting MAPK genes, derived from the global post-transcriptional network. The ERK module, consisting of *RAF1* (C-Raf), *MAP2K1* (MEK) and *MAPK1* (ERK2), is highlighted by red circles. *MAPK14* (p38α) is highlighted by a blue circle. (**B**) Real-time PCR validations of the RNA-sequencing data for some of the MAPK sub-network members, in SPHC and SPHV. Data are represented as mean ± SEM for n = 3 experiments. (**C**) ERK activity in SPHC and SPHV measured by Meso Scale Discovery (MSD) technology and expressed as the ratio of P-ERK/total ERK. Data are represented as mean ± SEM from n = 4 experiments. (**D**) Sprouting assay performed in the presence of the ERK inhibitor SHC 772984 or in the presence of the p38 inhibitor SB 202190, and the corresponding quantification of sprout area. Scale bars, 200 μm. ****, p<0.0001; ***, p<0.001; **, p<0.01; *, p<0.05; *ns*, not significant.

The online version of this article includes the following figure supplement(s) for figure 5:

**Figure supplement 1.** Assessment of the effective concentrations of ERK and p38 inhibitors in ECs.

transduced by non-targeting shRNA (*DICER*^WT) (*Figure 6—figure supplement 1A*), in agreement with previous observations (*Kuehbacher et al., 2007*). Effective *DICER* knock-down was verified by real-time PCR (*Figure 6—figure supplement 1B*). Then, we measured *RAF1*, *MAP2K1*, *MAPK1*, *TAOK1*, and *MAPK14* expression in SPHV generated with *DICER*^WT or *DICER*^KD cells by real-time PCR. All of these genes were upregulated in the *DICER*^KD condition (*Figure 6A*), indicating that their expression is subjected to a miRNA-mediated post-transcriptional control. The amount of P-ERK also increased in *DICER*^KD spheroids (*Figure 6B*). We repeated the sprouting assay with *DICER*^KD spheroids in the presence of the ERK inhibitor SCH 772984. *DICER*^WT spheroids were not affected by ERK inhibition, as shown previously in *Figure 5D*. Nonetheless, *DICER*^KD spheroids that were treated with the ERK inhibitor showed a further reduction of sprouting (*Figure 6C,D*), confirming that removal of miRNAs by *DICER* knock-down restored ERK pathway activity and thus sensitizing cells to the ERK inhibitor SCH 772984.

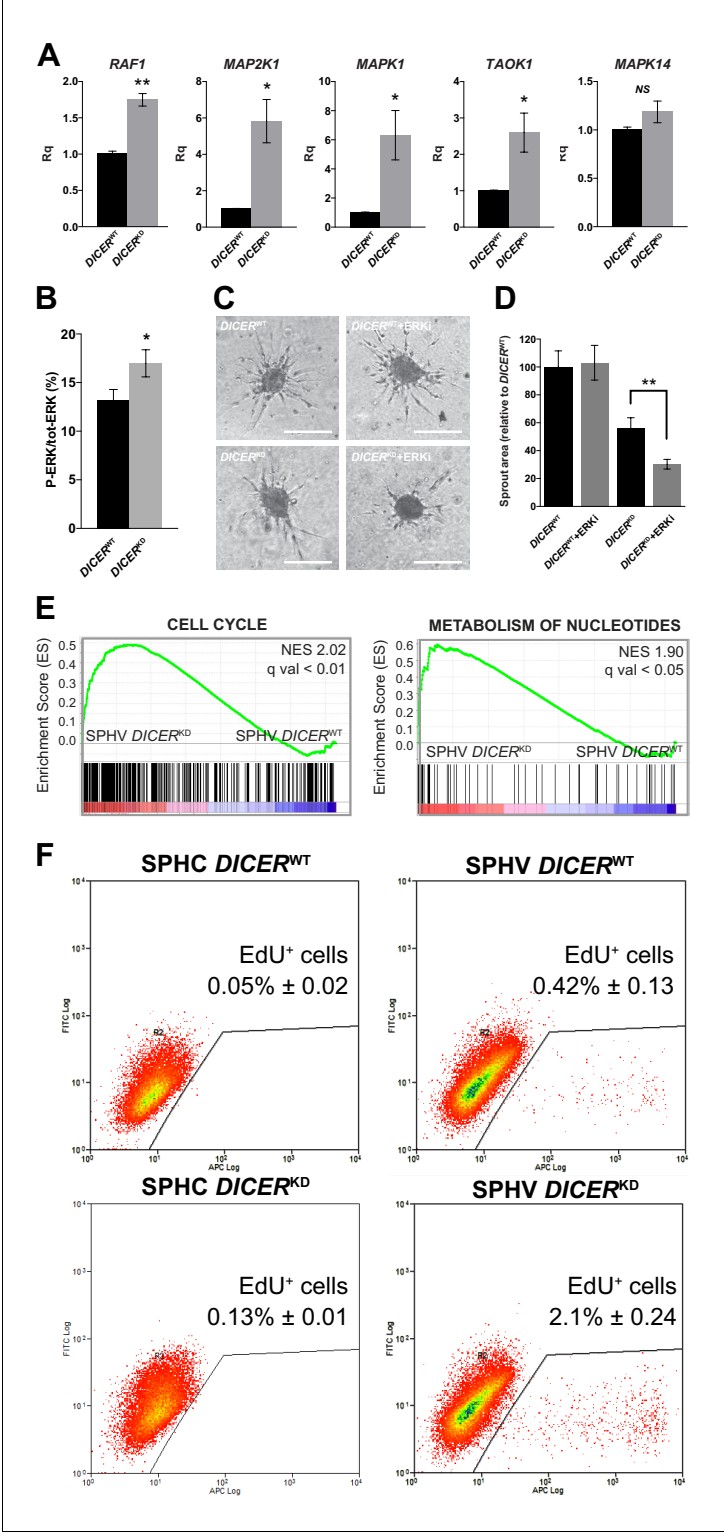

**Figure 6.** *DICER* knock-down rescues VEGF-A proliferative effect. (**A**) Real-time PCR analysis of *RAF1*, *MAP2K1*, *MAPK1*, *TAOK1*, and *MAPK14* in SPHV generated with *DICER*WT cells or *DICER*KD cells. Data are represented as mean ± SEM from n = 3 experiments. (**B**) ERK activity in SPHV generated with *DICER*WT cells or *DICER*KD cells measured by MSD and expressed as the ratio of P-ERK/total ERK. Data are represented as mean ± SEM from n = 3 experiments. (**C**) Sprouting assay performed with spheroids generated with *DICER*WT cells or *DICER*KD cells, in the presence of the ERK inhibitor SHC 772984. Scale bars, 200 μm. (**D**) Quantification of sprouts area for the

*Figure 6 continued on next page*

*Figure 6 continued*

cells shown in panel (**C**). Data are represented as mean ± SEM from n = 3 experiments. (**E**) GSEA study performed against the canonical pathways gene sets collection showed positive enrichment in the 'cell cycle' and 'metabolism of nucleotides' gene sets, in the comparison between *DICER*^KD and *DICER*^WT stimulated spheroids (SPHV). FDR was accepted when q value < 0.05. (**F**) Cell proliferation assessed by cytofluorimetric analysis of EdU incorporation into the DNA in spheroids generated from *DICER*^WT cells or *DICER*^KD cells that were either not stimulated (SPHC) or stimulated (SPHV) with VEGF-A. Representative plots of n = 3 experiments. Data represent mean percentages of proliferating cells ± SEM from n = 3 experiments. **, p<0.01; *, p<0.05.
The online version of this article includes the following figure supplement(s) for figure 6:

**Figure supplement 1.** *DICER* knock-down.

Furthermore, we performed a microarray study to analyze the gene expression profiles of *DICER*^WT and *DICER*^KD spheroids. GSEA comparison showed levels of expression of 'cell cycle'- and 'metabolism of nucleotides'-related genes in *DICER*^KD spheroids that were higher than those in *DICER*^WT spheroids stimulated with VEGF-A (*Figure 6E*), but no higher than those in control spheroids (*Figure 6—figure supplement 1C*). These findings indicate that the activity of miRNAs can restrain the expression of genes that promote cell proliferation following VEGF-A stimulation, in agreement with the network analysis (*Figure 3*) and the results shown in *Figure 2*. This observation was further validated by measuring proliferation rate in SPHC and SPHV (*Figure 6F*). *DICER*^KD spheroids displayed a VEGF-A-dependent increase of cell proliferation compared to *DICER*^WT spheroids, confirming that the removal of miRNAs allows activation of the proliferative pathway downstream of VEGF-A. Interestingly, when the proliferation assay was performed on ECs cultured in 2D, *DICER*^KD cells showed a reduced proliferative response to VEGF compared to *DICER*^WT cells, suggesting that different miRNA activities are involved in the 2D or 3D environments (*Figure 6—figure supplement 1D*).

## Biological validation of the miRNA hub network regulating sprouting angiogenesis

As previously shown, network analysis indicates that only few miRNAs are predicted to target a large number of protein-coding genes (*Figure 4—source data 1*), in agreement with the scale-free hypothesis that describes real networks. These hub miRNAs therefore should be pivotal to the network architecture and to the post-transcriptional regulatory activity that sustains the sprouting process. To validate the accuracy of the network analysis, and the relevance of the predicted miRNA-mediated post-transcriptional control of the genetic program sustaining SA, we subjected two miRNAs to further investigations. To validate the network, we selected miR-424–5p and miR-29a-3p, which are well-known miRNAs involved in angiogenesis regulation, and showed a high number of interactions with coding genes. miR-424–5 p was upregulated and targeted 98 genes, which were mainly related to intracellular signaling, including the MAPK pathway (*Figure 5A*) and cell cycle control (*Figure 7A* and *Figure 4—source data 1*). miR-29a-3p was downregulated and controlled the post-transcriptional regulation of 25 genes, mostly related to extracellular matrix remodeling (*Figure 7B* and *Figure 4—source data 1*). These miRNA–target interactions are also reported in the publicly available databases miRTarBase (*Chou et al., 2018*), TarBase (*Karagkouni et al., 2018*) and starBase (*Li et al., 2014*), which store miRNA–mRNA interactions supported by experimental evidence. In particular, for miR-424–5 p, 97 out of 98 interactions were present in these databases (24 of them in all three databases, and 73 in one or two); for miR-29a-3p, 23 out of 25 interactions were present in the databases (7 of them in all three databases, 16 in one or two) (*Figure 7—source data 1*). RNAseq expression data for miR-424–5p and miR-29a-3p in SPHC and SPHV were confirmed by real-time PCR analysis (*Figure 7—figure supplement 1A, B*). The expression of these two miRNAs can also be altered when the Notch pathway is manipulated. In fact, inhibition of the Notch pathway with DAPT, which increases sprouting (*Figure 1—figure supplement 2B*), increased the expression of miR-424–5p and reduced the expression of miR-29a-3p; the opposite effect was obtained when spheroids were treated with exogenous DLL4 (*Figure 7C,D*).

To assess the role of miR-424–5p and miR-29a-3p in the sprouting process, we altered their expression in ECs by using miRNA mimics or inhibitors (*Figure 7—figure supplement 1D,E*), and

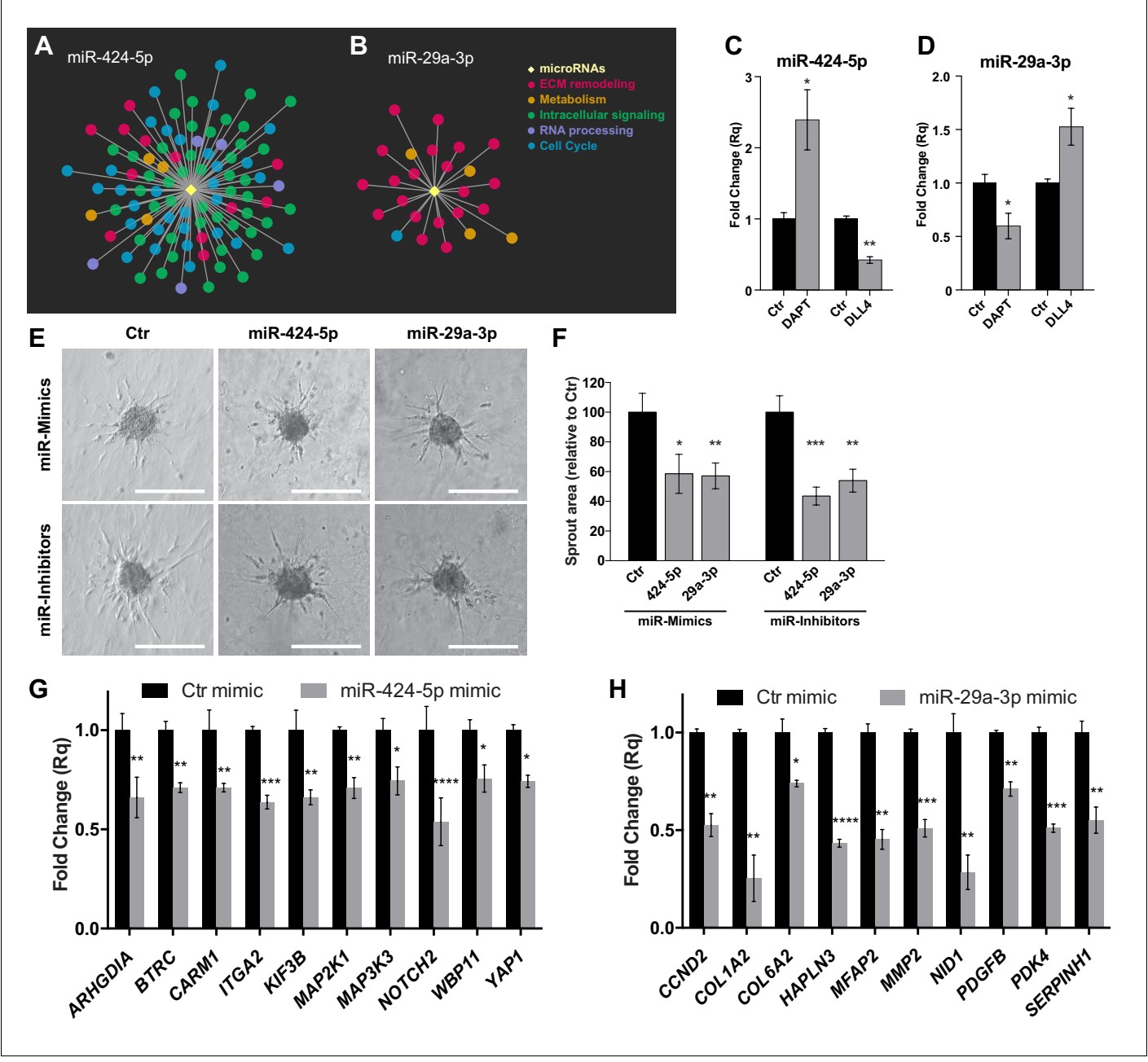

**Figure 7.** Hub miRNAs are essential to the sprouting process. Graphical representation of miR-424–5p (**A**) and miR-29a-3p (**B**) sub-networks, characterized by different patterns of functional classes target genes. (**C**) Real-time PCR analysis of miR-424–5p expression in SPHV treated with DAPT or with exogenous DLL4. Data are represented as mean ± SEM from n = 3 experiments. (**D**) Real-time PCR analysis of miR-29a-3p expression in SPHV treated with DAPT or with exogenous DLL4. Data are represented as mean ± SEM for n = 3 experiments. (**E**) Sprouting assay performed with ECs transfected with miR-424–5p or miR-29a-3p mimics and inhibitors. Scale bars, 200 μm. (**F**) Quantification of sprout area for the cells shown in panel (**E**). Data are represented as mean ± SEM from n = 4 experiments. (**G**) Real-time PCR assays in EC transfected with miR-424–5p mimic detecting the expression of miR-424–5p targets predicted from the network analysis. Data are represented as mean ± SEM from n = 3 experiments. (**H**) Real-time PCR assays in EC transfected with miR-29a-3p mimic detecting the expression of miR-29a-3p targets predicted from the network analysis. Data are represented as mean ± SEM from n = 3 experiments. ****, p<0.0001; ***, p<0.001; **, p<0.01; *, p<0.05.

The online version of this article includes the following source data and figure supplement(s) for figure 7:

**Source data 1.** Hub miRNA interactions supported by experimental evidence.

**Figure supplement 1.** miR-424–5p, miR-29a-3p and miR-16–5 p gain of function and loss of function.

**Figure supplement 2.** Tip competition assay.

**Figure supplement 3.** Assessment of VEGF-A-induced cell proliferation and migration upon modulation of miR-424–5p or miR-29a-3p.

*Figure 7 continued on next page*

*Figure 7 continued*

**Figure supplement 4.** Gene expression analysis of miR-424–5p and miR-29a-3p target genes upon miRNA inhibition.

performed the sprouting assay (*Figure 7E*). Quantification of sprout areas (*Figure 7F*) indicated that both upregulation by miRNA mimics and downregulation by miRNA inhibitors of miR-424–5p and miR-29a-3p impaired the ability of ECs to sprout, suggesting that the equilibrium of the network architecture requires that the expression of these two hub miRNAs is maintained at controlled levels. In a tip competition assay, spheroids were generated by mixing an equal number of ECs expressing the fluorescent protein DsRed with ECs transfected with miR-424–5p or miR-29a-3p mimics or inhibitors. Fluorescent or non-fluorescent cells occupying the tip position of the sprouts were counted. *Figure 7—figure supplement 2* shows that cells with altered expression of miR-424–5p or miR-29a-3p were impaired in reaching the tip cell position. To ascertain the robustness of the gain-of-function/loss-of-function approach, we also altered the expression of miR-16–5 p, which belongs to the same family as miRNA-424–5p, with which it putatively shares the same subset of target genes. However, our network analysis, which integrates target predictions with expression data, predicted a different subset of target genes for miR-424–5p and miR-16–5 p in the spheroid model (*Figure 4—source data 1*), and miR-16–5 p appeared to be a more peripheral miRNA that had only 16 predicted target genes. Indeed, miR-16–5 p modulation did not affect sprouting (*Figure 7—figure supplement 1F,G*). miR-424–5p and miR-29a-3p expression was significantly reduced in *DICER*[KD] spheroids compared to *DICER*[WT] spheroids (*Figure 7—figure supplement 1H*), suggesting that impaired expression of these two miRNAs could account, at least in part, for the sprouting defects observed in spheroids after *DICER* knock-down and in agreement with their role in network hierarchy. Furthermore, modulation of miR-424–5p, which is involved in the MAPK sub-network shown in *Figure 5A*, resulted in the alteration of P-ERK activity according to the bioinformatics prediction, whereas modulation of miR-29a-3p, which is not involved in this network, was not effective (*Figure 7—figure supplement 1I*).

The effect of modulation of miR-424–5p and miR-29a-3p on cell proliferation was also tested in the spheroid model, the limitation of this assay being that the inhibition of cell proliferation cannot be evaluated as it is already close to zero. Indeed, none of the treatments could significantly increase cell proliferation (*Figure 7—figure supplement 3A*). In a 2D VEGF-dependent cell proliferation assay, however, treatment with the miR-424–5p mimic was able to reduce cell proliferation by 50% (*Figure 7—figure supplement 3B*). When tested in a 2D VEGF-dependent cell migration experiment, the miR-424–5p mimic had a positive effect on cell migration, whereas the miR-29a-3p mimic had a negative effect, the latter in agreement with its predicted role as a negative regulator of cell migration. miRNA inhibitors, however, did not show any significant effect (*Figure 7—figure supplement 3C*).

To further validate miR-424–5p and miR-29a-3p activity in the different biological pathways identified by the network analysis, we used real-time PCR to analyze the expression of 10 predicted target genes for miR-424–5p and 10 for miR-29a-3p. The targets investigated included genes involved in cell proliferation control (*MAP2K1*, *CCND2*), as well as genes involved in cell migration and extracellular remodeling (*ITGA2*, *NID1*), whose alteration could account for the phenotype observed. Indeed, we were able to verify a reduction in the expression of target genes when the corresponding miRNA mimic was overexpressed (*Figure 7H,I*). The effect of the miRNA inhibitors on the selected targets was, however, less pronounced, and we were able to observe upregulation for only three target genes for each condition (*Figure 7—figure supplement 4A,B*). Therefore, we analyzed the expression of other miR-424–5p and miR-29a-3p targets that are present in their respective miRNA subnetworks and that were not previously analyzed. For most of these target genes, we observed a significant upregulation upon inhibition of miRNAs (*Figure 7—figure supplement 4C,D*).

## Modulation of miRNA hubs alters network architecture

The data showed above indicate, for some of the experiments performed, a non-symmetrical effect of miRNA upmodulation and downmodulation. This might be explained by the existence of a broader effect on the network architecture that would involve indirect miRNA–miRNA interactions. To gain more insights into the impact that modulation of a hub miRNA can have on the global

network architecture, we analyzed the expression of a large panel of miRNAs (of which 127 are represented in the SA network) by TaqMan array microRNA cards when miR-424–5p or miR-29a-3p were either upregulated or downregulated. The results indicate that 76 miRNAs showed altered expression ($|\Delta\Delta Ct| > 2$) upon up- or downmodulation of miR-424–5p or miR-29a-3p in at least two conditions (*Figure 8A*). In particular, when the analysis was restricted to the subset of miRNAs that share common target genes with miR-424–5p or miR-29a-3p in the SA network, 27 of these miRNAs showed altered expression (*Figure 8B*). Interestingly, several miRNAs showed increased expression under the effect of miR-424–5p or miR-29a-3p inhibitors. Altogether, these data indicate that that miR-424–5p and miR-29a-3p modulations induce a deep rewiring of the global miRNA network in addition to the modulation of their target coding genes, which can also account for the differences observed when comparing the effects of miRNA mimics and miRNA inhibitors.

## Upregulation of genes that are targeted by miRNAs during sprouting angiogenesis correlates with tumor angiogenesis

To extend the results derived from the SA model in a human setting, and to identify possible implications of our findings in tumor biology, we evaluated the expression of the global upregulated and downregulated gene modules extracted from the network analysis, as well as the expression of single miRNA targets, in a collection of colorectal cancers (CRC). This type of tumor was chosen because angiogenesis plays an important role in its progression (*Battaglin et al., 2018*), and some of the patients benefit from anti-angiogenic therapy with the VEGF blocker bevacizumab. 450 CRC samples were first ranked by their endothelial gene signature or endothelial score, a parameter that takes into account the expression of 35 EC-specific genes, not expressed by CRC or other stromal components, and that correlates with ongoing angiogenesis (*Isella et al., 2015*). When GSEA was used to correlate the different miRNA-target gene subsets obtained from the network analysis (*Figure 4—source data 1*) with CRC samples, we observed a significant positive enrichment for the miR-29a-3p-targets subset (*Figure 9A*), meaning that expression of miR-29a-3p targets is higher in CRCs that have a high endothelial score. Furthermore, average expression in CRC samples of the genes that were present in the network analysis and that are upregulated in SPHV (upregulated module, *Figure 9—source data 1*) also showed a significant positive correlation with endothelial score (*Figure 9B*). Likewise, GSEA analysis in which the endothelial score was used as phenotype classifier revealed a significant enrichment of the upregulated module (*Figure 9C*) and allowed the identification of a core gene set of 58 genes that are coherently regulated in vitro and in vivo (*Figure 9—source data 1*). This gene subset was further challenged in a dataset from CRC patients treated with the VEGF inhibitor bevacizumab (*Pentheroudakis et al., 2014*). Indeed, the core subset of the upregulated gene module was significantly enriched in bevacizumab-responding patients (*Figure 9D*), thus confirming the involvement of the genes within this module during pathological angiogenic remodeling.

## Discussion

### The spheroid model as a tool to study tip cell selection

In this study, we generated protein-coding gene and miRNAs RNAseq data from a VEGF-A dependent, well-established three-dimensional model of SA (*Korff and Augustin, 1998*; *Nowak-Sliwinska et al., 2018*). We showed that initial response to VEGF-A in ECs consists of a profound remodeling of the transcriptome, which allows differentiation toward the tip phenotype. This includes the activation of genes that regulate extracellular matrix remodeling, which is essential to the migratory process, and the repression of genes that promote cell proliferation and nucleotide de novo synthesis (*Figure 1B* and *Figure 2*). The observation that these two fundamental VEGF-A downstream pathways might not be synchronous in tip cells is in agreement with earlier studies on pathophysiological angiogenesis, in which it was demonstrated that the onset of vascular sprouting depends on the migration of existing ECs and does not require EC proliferation (*Ausprunk and Folkman, 1977*; *Sholley et al., 1984*). The spheroid model described here cannot, however, recapitulate all of the molecular events that lead to the formation of a more complex vascular bed. Furthermore, it cannot allow the analysis of tip-stalk cell dynamics because of the inability to differentiate stalk cells properly. This can be attributed to the intrinsic short life of the model

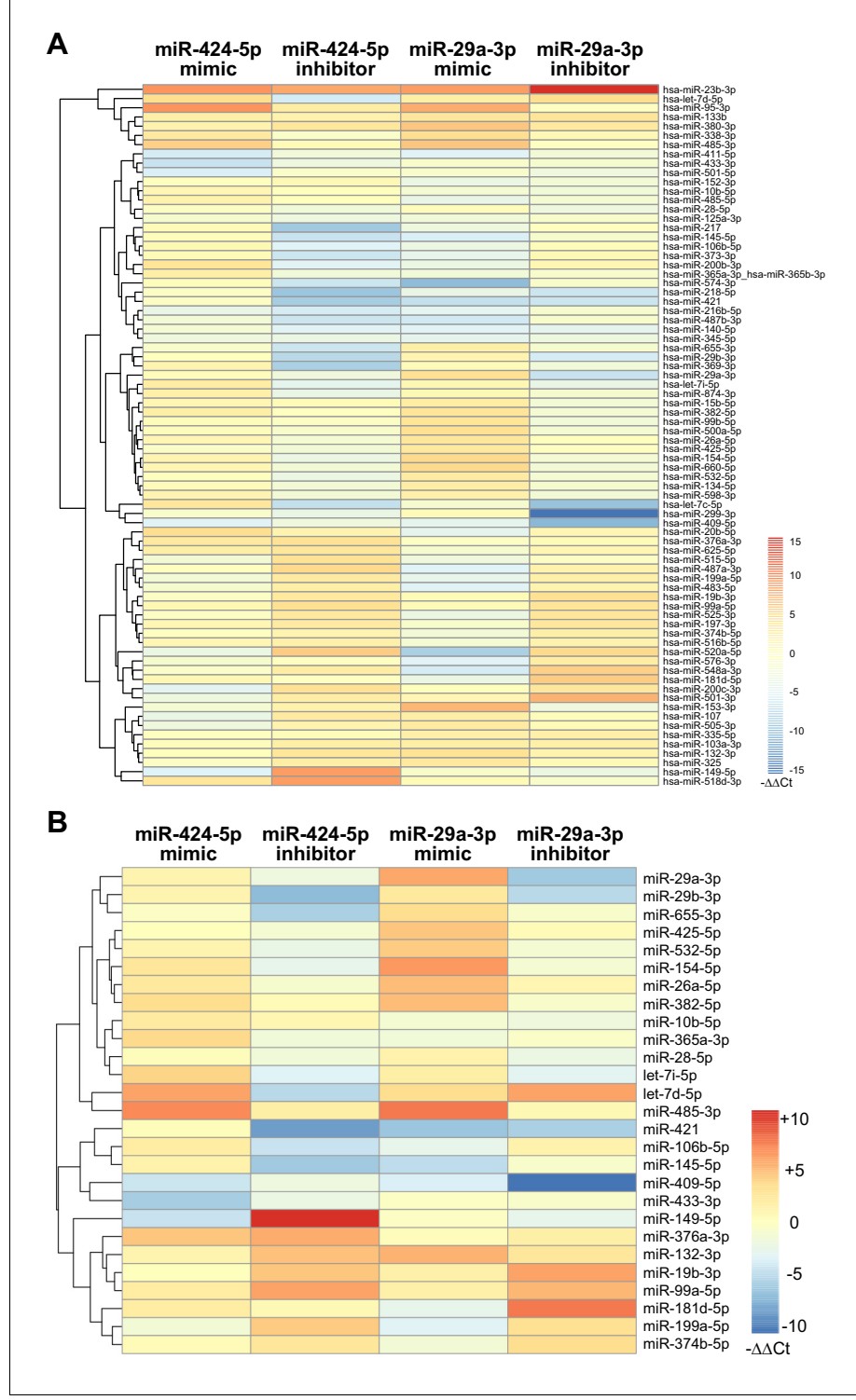

**Figure 8.** Modulation of miRNA hubs alters network architecture. (**A**) Heatmap showing the result of a differential gene expression analysis of a panel of miRNAs upon up- or downmodulation of miR-424–5p or miR-29a-3p. Only miRNAs with |ΔΔCt| > 2 are represented. Color bar indicates –ΔΔCt values. (**B**) Heatmap showing only those differentially expressed miRNAs that have predicted target genes in common with miR-424–5p or miR-29a-3p. Color bar indicates –ΔΔCt values.

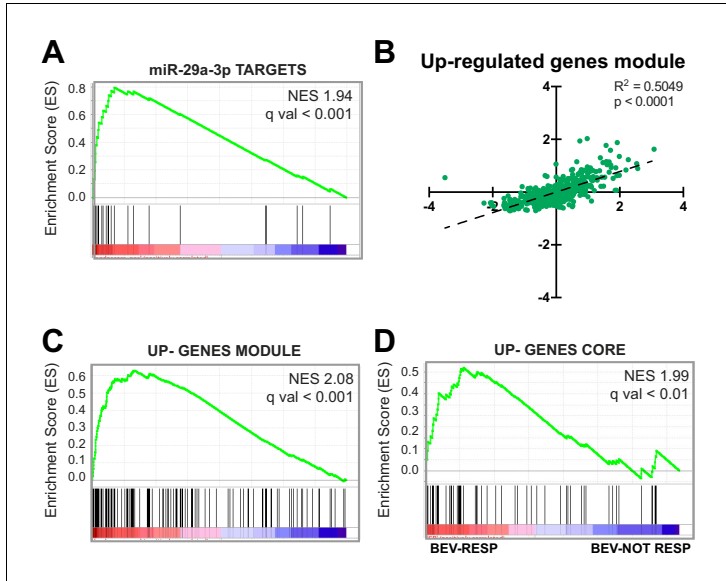

**Figure 9.** Upregulation of genes that are targeted by miRNAs during SA correlates with tumor angiogenesis. (**A**) GSEA plot showing significant positive association for miR-29a-3p target genes in CRC samples stratified by their endothelial score. (**B**) Scatter plot showing correlation, in CRC samples, between the expression of genes that constitute the upregulated gene module from the network analysis and endothelial score. (**C**) GSEA plot showing significant positive association for the up-gene module with CRC samples stratified by their endothelial score. (**D**) GSEA plot showing significant positive association for the upregulated genes core, extracted from panel (**B**), with a cohort of bevacizumab-responding CRC patients (bev-resp), compared to not-responding patients (bev-not resp). FDR was accepted when q value < 0.05.

The online version of this article includes the following source data for figure 9:

**Source data 1.** Genes constituting the upregulated gene module and the enrichment core in CRC.

compared to the mouse retina model and to the absence of other co-regulatory events, such as the generation of a VEGF gradient by neurons (*Okabe et al., 2014*). Nonetheless, the specificity of this model in the generation of tip cells allowed the analysis, at a molecular level, of the post-transcriptional regulation that sustains tip cells selection.

## Functions of miRNA in the context of regulatory networks

Here, information about the expression of miRNAs was used to generate a global post-transcriptional regulatory network based on co-expression between miRNAs and their predicted protein-coding target genes. Using this unbiased approach, we were able to emphasize a role of miRNAs in repressing genes that regulate cell proliferation and other signaling pathways downstream of VEGF. Such pathways include the MAPKs signaling cascade, which is mainly involved in transducing the VEGF-A mitogenic signal. Meanwhile, miRNAs also promoted the expression of genes that are involved in extracellular matrix remodeling.

Several studies have highlighted the importance of both transcriptional and post-transcriptional interactions in the shaping of global regulatory networks. The connections involving both transcription factors and miRNAs can be described in terms of small regulatory patterns (*Hornstein and Shomron, 2006*; *Tsang et al., 2007*). Some circuits topologies, in their incoherent form, can be associated with positive correlation patterns between the expression of miRNAs and their predicted target genes, and are generally thought to have a buffering function on transcriptional noise. On the other hand, the coherent form of miRNA-target circuits is thought to mediate a post-transcriptional reinforcement of transcriptional regulation and is associated with negative correlation schemes of gene expression (*Cora' et al., 2017*). Our experimental setup intrinsically highlights and selects such negative correlations (*Figure 3B*). In this context, miRNA activity is meant to reinforce the transcriptional program and to facilitate phenotypic transitions by precisely shaping a genetic program that is triggered by external cues (*Ebert and Sharp, 2012*). In principle, however, we cannot

preclude the presence, among the miRNA–gene network presented, of a later layer of indirect interactions between miRNAs and their targets or between the miRNAs themselves that is able to fine-tune the behavior of the network elements.

## Regulation of MAP kinase activity by miRNAs

To demonstrate carefully the robustness of the post-transcriptional regulatory network described here at a system level, we performed a series of experimental validation by envisaging two different approaches: 1) removal of mature miRNAs by *DICER* knock-down to evaluate the global effects of miRNA-mediated post-transcriptional activity on SA, and 2) validation of predicted targets for two miRNAs identified as hubs from the network analysis. For the first validation approach, we focused on a MAPK-specific subnetwork, due to its role in cell proliferation control, that pinpointed a coordinated miRNA inhibitory activity on the ERK module, represented by the *RAF1*, *MAP2K1* and *MAPK1* genes (*Figure 5A*). Removal of mature miRNAs by means of *DICER* knock-down reverted the downregulation of these genes that was observed after VEGF-A stimulus (*Figure 5B* and *Figure 6A*), increased P-ERK activity (*Figure 6B*), and restored a proliferative response (*Figure 6F*). Although the process that is triggered by a mitogenic stimulus depends on subsequent substrate phosphorylation throughout the MAPK cascade, it has been shown that the final outcome can also be influenced by miRNA-mediated post-transcriptional regulation of MAPK genes themselves. In particular, and in accordance with the interactions presented in this work, *RAF1* targeting by miR-497 inhibits breast cancer cell growth (*Li et al., 2011*), whereas downregulation of miR-424 leads to aberrant cell proliferation through MEK1 (*MAP2K1*) upregulation in senile hemangioma (*Nakashima et al., 2010*). Here, we also identified novel miRNA–MAPK interactions, such as miR-424–5p targeting *MAP2K1* and *MAP3K3*, or miR-19b-3p and miR-320b/c targeting *MAPK1*, although more specific experimental validations are required to demonstrate the function of these interactions.

Our data also indicate that the effect of miRNA activity in ECs is highly context-dependent. In fact, previous studies attributed the angiogenesis defects induced by *DICER* inhibition to a reduction of VEGF-induced proliferation (*Suárez et al., 2007*; *Suárez et al., 2008*). However, the mechanistic demonstration of this hypothesis was limited to in vitro experiments that used 2D-cultured ECs only. Although we confirmed the reported effects of *DICER* inhibition in 2D cultures, we also demonstrated that the activation of ECs that enables tip cell selection and migration (the first step in SA [*Carmeliet and Jain, 2011*]) benefits from a miRNA-mediated inhibition of cell proliferation. Therefore, in the more complex in vivo scenario, in which different ratios of migratory (tip) and proliferating (stalk) ECs exist simultaneously, miRNAs could have different effects depending on the EC phenotype considered. Nevertheless, owing to its broad effect on the expression of miRNAs, *DICER* inhibition could also have affected regulatory pathways other than ERK, and thus could have contributed to the observed phenotype.

## Role of two hub miRNAs in the post-transcriptional regulatory network

For the second validation approach, we focused on two hub miRNAs: miR-424–5p and miR-29a-3p, which belong to the network component containing up- and downregulated miRNAs, respectively. Their expression could be further modulated when the Notch pathway is either repressed or activated (*Figure 7C,D*), supporting the concept that their regulation promotes tip cell specification. Earlier studies in two-dimensional EC cultures reported that miR-424–5p upregulation reduced VEGF-induced proliferation (*Chamorro-Jorganes et al., 2011*), or indirectly modulated HIF-α expression thus promoting angiogenesis (*Ghosh et al., 2010*), whereas miR-29a inhibition impaired cell-cycle progression (*Yang et al., 2013*). However, it is also known that a widespread feature of miRNA activity toward a specific target is context-dependency, which also highlights the importance of target validation under physiological conditions (*Erhard et al., 2014*). In our three-dimensional model, integration of expression data with target prediction allowed the identification of context-specific targets. In fact, miR-424–5p showed preferential targeting activity towards genes that are associated with cell-cycle progression and intercellular signaling, whereas miR-29a-3p activity was more specific to genes involved in extracellular matrix remodeling (*Figure 7A,B*). A similar targeting pattern, for both miRNAs, has been described in previous evidence-based computational studies in different cellular models (*Tsang et al., 2010*). Our independent analyses in ECs support

the concept that a single miRNA can act on different targets belonging to the same pathway to reinforce its biological effect.

Finally, our unbiased approach disclosed an unpredicted hierarchical role of miR-424–5p and miR-29a-3p in a complex process such as SA. In fact, perturbation of miR-424–5p and miR-29a-3p expression levels propagated throughout the network, causing alterations in the expression levels of a large number of other miRNAs, including miRNAs that share coding targets with miR-424–5p or mir-29a-3p. Considering that each gene in the network is targeted by several different miRNAs, such broad network rearrangement could explain why inhibition of a single miRNA did not result in a significant increase in the expression of all its targets in a predictable way, and why inhibition of miR-424–5p alone was not as effective as removal of miRNAs by DICER knock-down in rescuing cell proliferation upon VEGF-A stimulus.

### Signature of the activity of miRNAs in tumor angiogenesis

To prove the strength of our experimental design and extend our findings to clinical data, we investigated the angiogenic post-transcriptional activity in human tumors. Indeed, in a collection of 450 CRC, we successfully challenged the correlation between gene modules that constitute a signature of the activity of miRNAs and ongoing angiogenesis in the stromal compartment. A significant correlation was, however, not detected for the downregulated gene modules, which mostly included cell-proliferation-associated genes. This can be explained by the fact that, in whole tumor transcriptomic data, the signal from the highly proliferating epithelial cancer cells overcome the signal coming from the less represented ECs. In addition, a signature composed of 58 sprouting-associated genes, which also included miR-29a-3p targets, was able to identify patients who benefitted from anti-angiogenic treatment with the VEGF-A blocking agent bevacizumab. On one side, this strengthens the evidence of the involvement of post-transcriptional regulation in tumor angiogenesis, and on the other side, it paves the way for future development of predictive markers for VEGF-A blocking agents. This analysis could also be extended to other cancer types, such as ovarian, cervical, non-small cell lung cancer or glioblastoma, in which angiogenesis is recognized to support tumor growth and in which bevacizumab is an approved therapeutic option. In summary, network analysis of whole transcriptomic data obtained from an in vitro SA model identified miRNA-mediated regulatory modules that fine-tune the response of ECs to VEGF-A and that could be exploited to identify new druggable biomarkers in angiogenesis-related diseases.

## Materials and methods

### Primary cell cultures

Human umbilical vein endothelial cells (HUVECs) were isolated from the cords of newborns by collagenase digestion of the interior of the umbilical vein, as previously described (*Nowak-Sliwinska et al., 2018*). To reduce the experimental variability that results from the different genetic backgrounds of individuals, each HUVEC batch was composed of cells derived from three to five different cords. HUVECs were cultured in M199 medium supplemented with 20% fetal bovine serum (FBS), 0.05 µg/ml porcine heparin, 2% penicillin-streptomycin solution (all from Sigma-Aldrich, St. Louis, MO, USA) and 0.2% brain extract. HUVECs were used for experiments until the third passage. Collection of umbilical cords for the isolation of HUVECs is governed by an agreement between Università degli Studi di Torino and the Azienda Ospedaliera Ordine Mauriziano di Torino, protocol number 1431 02/09/2014. Informed consent was obtained from all subjects involved.

### Spheroid capillary sprouting assay

EC spheroids were generated as described previously (*Nowak-Sliwinska et al., 2018*) with minor modifications. HUVECs within the third passage were trypsinized and cultured in hanging drops (800 cells/drop) in M199 containing 10% FBS and 0.4% (w/v) methylcellulose (Sigma-Aldrich). After 10 hr or overnight incubation, spheroids were collected and embedded in a solution containing 15% FBS, 0.5% (w/v) methylcellulose, 1 mg/ml rat tail collagen I solution, 30 Mm HEPES and M199 from 10X concentrate (all from Sigma-Aldrich). 0.1M NaOH was added to adjust the pH to 7.4 to induce collagen polymerization. After 30 min incubation at 37°C, polymerized gel was overlaid with M199 medium. Sprouting was induced by addition of 20 ng/ml recombinant human VEGF-A (R and D

Systems, Minneapolis, MN, USA) to the collagen solution and to the overlaying medium. After 18 hr incubation at 37°C in a 5% $CO_2$ incubator, spheroids were imaged, or collected for further analysis. When indicated, inactivation of cell proliferation was performed by treating the cells with mitomycin C (Sigma-Aldrich) at 10 μg/ml for 2 hr in M199, 24 hr prior spheroids generation. When spheroids assay was performed in the presence of drugs, the drugs were added at the moment of collagen I embedding. The fluorescent ECs used in the tip competition assay were generated by transduction with lentiviral particles carrying the plasmid pLVX-DsRed-Express2-N1 (Takara, Mountain View, CA, USA) followed by puromycin selection for 48 hr. For experiment of Notch pathway modulation, spheroids were incubated with either DAPT (Sigma-Aldrich) 2 μM or human recombinant DLL4 (R and D Systems) 1 mg/ml.

## Imaging

Spheroids in 3D collagen matrix were fixed with 4% PFA for 30 min at room temperature and imaged under bright field using an AF6000LX-TIRF workstation (Leica, Wetzlar, Germany). Measurement of the total sprout area of individual spheroids was performed using the Image J software package. For each experimental condition, the sprout area of at least 20 spheroids was measured and averaged to achieve 5% significance and 80% power.

## RNA isolation

To isolate ECs after the sprouting assay, the collagen matrix was digested by incubation with 0.25% Collagenase A (Sigma-Aldrich) in M199 medium at 37°C for 10 min. Total RNA was isolated from HUVECs by using TRIzol (ThermoFisher Scientific, Waltham, MA, USA) reagent and an miRNeasy Mini Kit (Qiagen, Hilden, Germany). Residual DNA was removed by treatment with Rnase-Free Dnase Set (Qiagen). The quality and concentration of RNAs were assessed with a NanoDrop ND-1000 spectrophotometer (ThermoFisher Scientific). The quality of the RNA samples that were subsequently processed for RNA-sequencing and microarray analysis was further assessed using a Qubit RNA HS Assay Kit and an Agilent RNA 6000 Nano Kit in an Agilent Bioanalyzer (Agilent, Santa Clara, CA, USA).

## RNA-sequencing analysis

Total RNA from spheroids exposed to VEGF-A for 18 hr (SPHV) and control spheroids (SPHC) that were not exposed to VEGF-A were subjected to high throughput sequencing for poly-A + RNAs by an external next generation sequencing (NGS) facility (Fasteris, Geneva, Switzerland). Three biological replicates from each biological condition were analyzed. RNA sequencing was performed on a HiSeq 2000 sequencer (Illumina, San Diego, CA, USA), obtaining a mean of 100 million paired-end reads of 100 bps per sample. Reads to counts conversion and differential expression analysis were performed using the software implementation described by *Beccuti et al. (2018)*. In brief, fastq files were mapped with STAR-2.5 to the reference genome (hg19 assembly) and gene counts were generated using RSEM-1.3.0 and ENSEMBL annotation. Differential expression analysis was performed using DESeq2 version 1.14.1 (*Love et al., 2014*) to detect modulated genes (| $\log_2$FC |>0.5 and FDR < 0.05). Data were deposited in the GEO database, with accession number GSE115817.

For miRNAs, the same external NGS facility (Fasteris, Geneva, Switzerland) was used to perform small-RNA sequencing profiling on the same matched samples as those used for mRNA sequencing. Sequencing was performed on a HiSeq 2000 sequencer (Illumina), obtaining a mean of 8 million 50-bps single-end reads per sample. From the raw data, adaptors were first trimmed by the FASTX-Toolkit. Then, inserts were mapped by the BWA tool (*Li and Durbin, 2010*) on mature miRNA sequences from miRBase v22 (*Kozomara and Griffiths-Jones, 2014*), thus producing an expression level for all annotated miRNAs. Data were deposited in the GEO database, with accession number GSE115954.

## Microarray analysis

450 ng total RNA was employed to synthetize biotinylated cRNA probes using an Illumina TotalPrep RNA Amplification Kit (ThermoFisher Scientific), according to the manufacturer's instructions. After quantification in an Agilent Bioanalyzer, 750 ng cRNAs were hybridized to an Illumina HumanHT-12

v4.0 Expression Bead Chip for 18 hr at 58°C. Biotinylated probes were then labeled with streptavidin-Cy3. Fluorescent signal was acquired by an Illumina BeadArray Reader, and data were extracted using the GenomeStudio software V2011.01. Probe intensities across the chip were normalized by applying the cubic spline normalization algorithm. For subsequent analysis, only probes with a detection p-value <0.05 were selected. For each gene, we retained the associated probe with the largest mean expression value across all samples. For each probe, the $\log_2$ signal was converted to the $\log_2$ ratio against the global average expression of that probe in all samples. Each experimental point was assessed with two biological replicates. Data were deposited in the GEO database, with accession numbersGSE116039 and GSE129275.

## Gene set enrichment analysis (GSEA)

GSEA was run on the dataset of protein-coding genes in SPHV versus SPHC against the canonical pathway genesets collection (c2.cp.v5.2, Broad Institute) by using 1000 genesets permutations. Only genesets with FDR < 0.05 were considered. Meta-analysis on colorectal cancer was performed on RNAseq data provided by The Cancer Genome Atlas (TCGA) (*Cancer Genome Atlas Network, 2012*), and on microarray data from the GEO Series GSE53127. GSEA was performed by evaluating the enrichment of upregulated modules and miRNA targets for genes that had high correlation with endothelial scores. Only genesets with FDR < 0.05 were considered.

## Generation of the post-transcriptional network

Pair-wise correlation analysis was performed on the gene expression profiles of protein-coding genes and miRNAs from the RNA-sequencing data of SPHC and SPHV samples by using Pearson correlation as co-expression measure. Association between functional biological pathways and miRNAs was performed using the GSEA algorithm. Briefly, miRNA expression data were used as the phenotype to calculate the Pearson correlation coefficient between each pair of miRNA and protein-coding genes and to find association with biological processes. Genesets with FDR < 0.05 that positively or negatively correlated with miRNAs expression were extracted and used to generate a matrix that was subsequently clusterized using GenePattern (Broad Institute). Genes constituting the core enrichment were extracted from significant genesets and subjected to functional annotation using the online David functional annotation platform. To find associations between miRNA and their protein-coding target genes, we extracted data from TargetScan database V7.1, selecting evolutionarily conserved interactions with a weighted context++ score percentile >50. For non-conserved interactions, we considered a weighted context++ score percentile >90. Graphical representation of networks was performed using Cytoscape software, version 3.3.0 (*Shannon et al., 2003*). Power-law fitting and the corresponding statistical analysis were performed in MATLAB, according to *Clauset et al. (2009)*.

## Real-time PCR

To analyze mRNAs expression, 1 μg of DNAse-treated RNA was reverse-transcribed using a High Capacity cDNA Reverse Transcription kit and random primers (ThermoFisher Scientific). cDNA amplification was performed using gene-specific TaqMan assays and TaqMan PCR Universal MasterMix (ThermoFisher Scientific) in a CFX96 thermocycler (Bio-Rad, Hercules, CA, USA). Each assay was run in triplicate. To analyze miRNA expression, 350 ng of DNAse-treated RNA was reverse-transcribed using a High Capacity cDNA Reverse Transcription kit in the presence of a miRNA-specific primer (ThermoFisher Scientific). Real-time PCR was performed using miRNA-specific TaqMan MicroRNA Assays and TaqMan Universal PCR Master Mix (ThermoFisher Scientific) in a CFX96 (Bio-Rad) Real-Time PCR. Each assay was run in triplicate. Expression of *TBP* or *RNU44* was used as the endogenous control for mRNA or miRNA relative quantifications, respectively. Relative quantification of gene expression levels between samples (Rq) was performed using the comparative Ct (threshold cycle number) method (*Livak and Schmittgen, 2001*). A complete list of the Real-time PCR assays used is provided in *Supplementary file 2*.

miRNA expression profiling under miRNA hubs modulation was performed using TaqMan Advanced miRNA Human A Card (ThermoFisher Scientific). 10 ng of total RNA from each sample were retrotranscribed using a TaqMan Advanced miRNA cDNA Synthesis Kit (ThermoFisher Scientific). Each sample was run in biological duplicate. Expression data were first normalized following

manufacturer's instructions; subsequently, the ΔΔCt method was used to evaluate the relative expression level (fold change) in each condition versus the corresponding control. ΔΔCt values were loaded in the R environment software for statistical analysis and the heatmap function was used to perform hierarchical clustering analysis and to represent data in a heatmap.

## Transient transfections

Sub-confluent ECs were transfected with the mirVana miRNA mimic, to overexpress miRNAs, or with mirVana miRNA inhibitor (ThermoFisher Scientific), to inhibit miRNAs activity, at a final concentration of 90 nmol/L by using RNAiMAX lipofectamine in Optimem medium (ThermoFisher Scientific), according to the manufacturer protocol. Effective overexpression or inhibition was verified by real-time PCR, as described. Subsequent assays were performed 24 hr post-transfection.

## shRNA-mediated gene knock-down

*DICER1* knock-down was performed by transducing ECs with shRNAs carried by lentiviral particles. Self-inactivating, replication-incompetent lentiviral particles were generated by cotransfection of expression plasmid (TRC1 pLKO.1-puro Non-Target shRNA as negative control or TRC1 pLKO.1-puro carrying *DICER1*-targeting shRNAs, Sigma-Aldrich), envelope plasmid (pVSV-G) and packaging plasmid (pCMV-Dr8.91) into HEK-293t cells using the calcium phosphate method, as previously described (*Follenzi et al., 2000*). ECs were transduced with lentiviral particles at a multiplicity of infection (MOI) = 1 in the presence of 8 μg/ml polybrene. Medium was replaced after 24 hr, and cells stably expressing the shRNA were selected by growth in puromycin (1 μg/ml) for 24 hr. Effective *DICER* knock-down of two different shRNAs was evaluated by real-time PCR. The shRNA with the highest silencing efficacy was chosen (sh#3) and used for subsequent experiments. Spheroid assays were performed 48 hr post-transduction.

## Western blot analysis

Total proteins were obtained by cell lysis in an extraction buffer containing 0.125 M Tris-HCl (pH 6.8), 4% SDS, 20% glycerol, and quantified by BCA assay (ThermoFisher Scientific). Equal amounts of protein extract per sample were separated on an SDS-PAGE and subsequently blotted onto a PVDF membrane. Membranes were incubated with specific primary antibodies and appropriate HRP-conjugated secondary antibodies. Immunoreactive proteins were visualized using an enhanced chemiluminescence (ECL) system and acquired using a ChemiDoc Touch Gel Imaging System (Biorad). Images were analyzed with Image Lab software 5.2.1 (Biorad). Antibodies used were anti-VEGFR2, anti-P38, anti-phospho-P38, anti-ERK1/2, anti-phospho-ERK1/2 (Cell Signaling Technology, Danvers, MA, USA), and anti-GAPDH (Abcam, Cambridge, UK).

## ERK activity assessment

Total ERK and phospho-ERK proteins were measured using the Meso Scale Discovery (MSD) technology. MAP Kinase (Total Protein) Whole Cell Lysate Kit and MAP Kinase Whole Cell Lysate Kit MULTI-SPOT plates were used (Meso Scale Diagnostics, Rockville, MA, USA) according to the MSD manufacturer's information. 7 μg of each sample were used for the detection. The amount of phosphoprotein was calculated according to the MSD directions using the following formula: % phosphoprotein = ((2 x phospho-signal) / (phospho-signal + total signal)) x 100.

## Whole-mount immunofluorescent staining

Collagen gels containing spheroids were fixed with 4% paraformaldehyde for 30' min, followed by 3 hr incubation in blocking buffer (PBS 0.5% Triton + 0.1% tween 20% and 10% donkey serum). Anti-DLL4 antibody (Abcam) or anti-CXCR4 antibody (Abcam) were diluted 1:1000 in blocking buffer and incubated overnight at 4°C. After extensive washing in blocking buffer, collagen gels were incubated with AF555-conjugated secondary antibody (ThermoFisher Scientific) in the same buffer for 1 hr. Cell nuclei were counterstained with DAPI.

## Notch signaling activity

The plasmid pWPT-12XCSL-DsRedExpressDR was generated by subcloning the Notch fluorescent reporter 12xCSL-DsRedExpressDR (*Hansson et al., 2006*) into the lentiviral vector pWPT.

Generation of lentiviral particles and transduction of ECs were performed as described above. Spheroid assays were performed 48 hr post-transduction.

## Proliferation assay

A proliferation assay was performed using Click-It EdU Flow Cytometry Cell Proliferation Assay (ThermoFisher Scientific). For 2D proliferation assays, $2.5 \times 10^3$ ECs were plated in six well plates and starved in serum-free M199 supplemented with 0.1% BSA for 10 hr. Cell proliferation was induced by the addition of 20 ng/ml VEGF-A to the culture medium, and maintained for additional 18 hr. 10 μM of 5-ethynyl-2′-deoxyuridine (EdU) was added 2 hr prior to cell harvesting. For cell proliferation measurement in spheroids, 10 μM EdU was added 2 hr prior to spheroids collection. EdU detection was performed according to the manufacturer's protocol and samples were analyzed in a CyAn ADP flow cytometer with Summit five software (Beckman Coulter, Brea, CA, USA).

## Migration assay

Real-time directional cell migration was measured with an xCELLigence RTCA DP instrument in a CIM plate (ACEA Biosciences, San Diego, CA, USA). The bottom of the upper chamber was coated with 1% gelatin for 5 min at room temperature. The lower chamber was filled with M199 medium, 10% FBS, and VEGF 20 ng/ml as chemoattractant. $3 \times 10^5$ HUVECs were seeded in the upper chamber in 100 μl M199 medium. ECs migration was continuously monitored by measuring the Cell Index every 30 min for 24 hr in a $CO_2$ incubator at 37°C. Data were analyzed with the xCELLigence RTCA software (ACEA Biosciences). The slope of the curve obtained by Cell Index measurements, which represents the speed of cell migration, was calculated the slope.

## Enzymatic activity measurements

### Purine synthesis

The activity of glutamine amidophosphoribosyltransferase (PPAT) was measured as described previously (*Capello et al., 2016*). The [$^{14}$C]-L-glutamate generated from the reaction was separated from [$^{14}$C]-L-glutamine by ion exchange chromatography in a 2 ml column. The radioactivity of the eluate containing [$^{14}$C]-L-glutamate was counted by liquid scintillation and expressed as nmol glutamate (Glu)/h/mg cell proteins. The activity of aminoimidazole-4-carboxamide ribonucleotide formyltransferase/inosine monophosphate cyclohydrolase (ATIC) was measured as reported previously (*Boccalatte et al., 2009*). Results were expressed as nmol NADP+/min/mg cell proteins.

### Pyrimidine synthesis

The activity of carbamoyl phosphate synthetase II (CAD) was measured as described previously (*Capello et al., 2016*). Results were expressed as pmol carbamoyl aspartate (CA)/min/mg cell proteins.

## ERK and p38 pharmacological inhibition

$6 \times 10^5$ cells were plated in six well plates and pretreated with the ERK inhibitor SHC772984 (Cayman Chemical, Ann Arbor, MI, USA) or the p38 inhibitor SB 202190 (Sigma-Aldrich) for 18 hr, using the following concentrations: 30, 100 or 300 nM for ERK inhibitor, and 10, 100 or 300 μM for p38 inhibitor. Cells were starved for 3 hr in serum-free M199 and then stimulated with 20 ng/ml VEGF-A for 10 min. Starving and stimulation were performed in the presence of the drugs. Cells were then processed for protein extraction and western blot analysis.

## Statistical analyses

All statistical analyses were performed using the GraphPad Prism six software, the R statistical package, or MATLAB. For all the datasets included in t-tests, Shapiro-Wilk normality test was applied to assess normal distribution. Pooled data are expressed as the mean ± SEM. n represents the number of biological replicates, each performed with different HUVEC batches. Significance was determined by using unpaired Student's t-test (two tailed); $p < 0.05$ was considered significant. Specific details for each experiment can be found in the corresponding figure legend. Elsewhere, for hypergeometric tests, $p < 0.05$ was considered significant; for the Kolmogorov-Smirnov test, $P > 0.1$ was considered significant.

## Data access

All raw and processed sequencing data generated in this study have been submitted to the NCBI Gene Expression Omnibus (GEO; http://www.ncbi.nlm.nih.gov/geo/) and are available under the accession numbers GSE116039, GSE115954, GSE115817 and GSE129276.

# Acknowledgements

This work was supported by AIRC – Associazione Italiana Per la Ricerca sul Cancro (grants 22910, 12182 and 18652), Regione Piemonte (grant A1907A, Deflect), Fondazione CRT, Ministero dell'Università e della Ricerca (PRIN 2017, grant 2017237P5X), FPRC 5xmille 2016 MIUR (Biofilm) and ERA-Net Transcan-2 (grant TRS-2018–00000689) to FB. We thank Prof. Urban Lendahl for the gift of the Notch reporter plasmid and Dr. Jeffrey F Thompson for revising the manuscript.

# Additional information

### Funding

| Funder | Grant reference number | Author |
| --- | --- | --- |
| Associazione Italiana per la Ricerca sul Cancro | 12182 | Federico Bussolino |
| Associazione Italiana per la Ricerca sul Cancro | 18652 | Federico Bussolino |
| Regione Piemonte | A1907A Deflect | Federico Bussolino |
| Ministero dell'Istruzione, dell'Università e della Ricerca | PRIN 2017, 2017237P5X | Federico Bussolino |
| Ministero della Salute | Era-net TRS-2018-00000689 | Federico Bussolino |
| Fondazione CRT | | Federico Bussolino |
| Associazione Italiana per la Ricerca sul Cancro | 22910 | Federico Bussolino |
| Ministero dell'Istruzione, dell'Università e della Ricerca | FPRC 5xmille 2016 Biofilm | Federico Bussolino |

The funders had no role in study design, data collection and interpretation, or the decision to submit the work for publication.

### Author contributions

Stefania Rosano, Conceptualization, Validation, Investigation, Methodology; Davide Corà, Data curation, Formal analysis; Sushant Parab, Formal analysis, Bioinformatics analyses on publicly available databases; Serena Zaffuto, Chiara Riganti, Validation, Investigation; Claudio Isella, Formal analysis, Methodology; Roberta Porporato, Roxana Maria Hoza, Investigation, Methodology; Raffaele A Calogero, Software, Formal analysis; Federico Bussolino, Conceptualization, Supervision, Funding acquisition; Alessio Noghero, Conceptualization, Formal analysis, Supervision, Validation, Investigation, Visualization, Methodology

### Author ORCIDs

Alessio Noghero (iD) https://orcid.org/0000-0002-8266-9247

### Ethics

Human subjects: Informed consent was obtained from all subjects involved. Collection of umbilical cords is governed by an agreement between Università degli Studi di Torino and the Azienda Ospedaliera Ordine Mauriziano di Torino, protocol number 1431 02/09/2014.

### Decision letter and Author response

Decision letter https://doi.org/10.7554/eLife.48095.sa1

Author response https://doi.org/10.7554/eLife.48095.sa2

## Additional files

### Supplementary files

- Supplementary file 1. Key resources table.
- Supplementary file 2. Real-time PCR assays list.
- Transparent reporting form

### Data availability

Sequencing data have been deposited in GEO under accession codes GSE116039, GSE115954, GSE115817, GSE129276.

The following datasets were generated:

| Author(s) | Year | Dataset title | Dataset URL | Database and Identifier |
|---|---|---|---|---|
| Noghero A, Busso-lino F, Corà D, Rosano S | 2019 | A Regulatory microRNA Network Controls Endothelial Cell Phenotypic Switch During Sprouting Angiogenesis | https://www.ncbi.nlm.nih.gov/geo/query/acc.cgi?acc=GSE116039 | NCBI Gene Expression Omnibus, GSE116039 |
| Noghero A, Busso-lino F, Corà D, Rosano S | 2019 | A Regulatory microRNA Network Controls Endothelial Cell Phenotypic Switch During Sprouting Angiogenesis | https://www.ncbi.nlm.nih.gov/geo/query/acc.cgi?acc=GSE115954 | NCBI Gene Expression Omnibus, GSE115954 |
| Noghero A, Busso-lino F, Corà D, Rosano S | 2019 | A Regulatory microRNA Network Controls Endothelial Cell Phenotypic Switch During Sprouting Angiogenesis | https://www.ncbi.nlm.nih.gov/geo/query/acc.cgi?acc=GSE115817 | NCBI Gene Expression Omnibus, GSE115817 |
| Noghero A, Busso-lino F, Corà D, Rosano S | 2019 | A Regulatory microRNA Network Controls Endothelial Cell Phenotypic Switch During Sprouting Angiogenesis | https://www.ncbi.nlm.nih.gov/geo/query/acc.cgi?acc=GSE129276 | NCBI Gene Expression Omnibus, GSE129276 |

The following previously published dataset was used:

| Author(s) | Year | Dataset title | Dataset URL | Database and Identifier |
|---|---|---|---|---|
| Pentheroudakis G, Kotoula V, Fountzi-las E, Kouvatseas G, Basdanis G, Xanthakis I, Makat-soris T, Charalam-bous E, Papamichael D, Samantas E, Papa-kostas P, Dimitrios B, Razis E, Chris-todoulou C, Varthalitis I, Fount-zilas G | 2013 | Study of gene expression markers for predictive significance for bevacizumab benefit in patients with metastatic colon cancer: A translational research study of the Hellenic Cooperative Oncology Group (HeCOG) | https://www.ncbi.nlm.nih.gov/geo/query/acc.cgi?acc=GSE53127 | NCBI Gene Expression Omnibus, GSE53127 |

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
