## [Decision Letter]

**Acceptance summary:**

MicroRNAs (miRNAs) are pivotal regulator of gene expression during cell behaviors. Yet, the network of target-gene regulation that enables their mode of function is often difficult to decipher in biological systems due to the high complexity of their predicted mRNA matches. Sprouting angiogenesis is mediated by the activation of the vascular endothelial growth factor (VEGF-A) signaling cascade that coordinate proliferation, migration and survival of endothelial cells. Using a combination of computational and genomics analysis of VEGF-signaling responses in 3D spheroid and 2 D endothelial cell-based assays, the authors show that VEGF-A activated a complex miRNA-centered co-expression network characterized by the repression of cell cycle genes and de-repression of extracellular matrix remodeling and cell migration genes. The consequences of loss of miRNA processing, via Dicer silencing, support that VEGF-A activates such network changes to enable migration instead of a proliferation phenotype during tip-cell formation in the sprouting angiogenesis of the 3D-spheroid model. In addition, miRNAs were found dispensable for VEGF-A-mediated angiogenesis in the 2D endothelia cell model. This work illuminates the high degree of post-transcriptional regulation that controls endothelial cells responses to VEGF-A and will improve our understanding of different modes of VEGF-A function during the genesis of blood vessels in diverse tissues.

**Decision letter after peer review:**

Thank you for submitting your article "A regulatory microRNA network controls endothelial cell phenotypic switch during sprouting angiogenesis" for consideration by *eLife*. Your article has been reviewed by three peer reviewers and the evaluation has been overseen by Didier Stainier as the Senior Editor. The following individual involved in review of your submission has agreed to reveal their identity: Minna U Kaikkonen (Reviewer #2).

The reviewers have discussed the reviews with one another and the Reviewing Editor has drafted this decision to help you prepare a revised submission.

Summary:

The manuscript by Rosano et al. investigates the role of miRNAs in the regulation of angiogenesis in 3D in vitro model. They compare the 3D angiogenesis model to 2D angiogenesis and demonstrate that the 3D conditions directs cell migration over cell proliferation. In the second part of the manuscript, they identify post-transcriptional regulatory networks by performing pair-wise correlation of mRNA and miRNA expression. They provide evidence that MAPKs play an important role in sprouting angiogenesis through regulation of RAF1, MAP2K1, MAPK1 and TAOK1 expression. Finally, they provide characterization of the effect of miR-424-5p and miR-29a-3p in angiogenesis and regulation of genes implicated in cell proliferation and migration.

General assessment:

The reviewers find that the paper describes a very interesting study of the mechanism by which VEGF-A-signaling coordinate tip cells behaviors during sprouting angiogenesis via controlling post-transcriptional regulation.

Central conclusions:

1) Trascriptome analysis indicates that a proliferation-to-migration switch occurs during VEGF-A induction of sprouting angiogenesis in 3D spheroids vs. 2D cell culture assay.

2) Post-transcriptional regulatory networks driven by the miRNA machinery protein, Dicer, are essential to suppress proliferative tip cells behavior in 3D and promote motile behavior.

3) miR-424-5p and miR-29a-3p regulation of genes hub is implicated in buffering cell proliferation and migration during sprouting angiogenesis.

The reviewers raise a number of concerns that must be addressed before the paper can be accepted. There is some skepticism on how the data discovered with the spheroids 3D vs. 2D models can be relevant for the scientific community that study sprouting angiogenesis in vivo. This point could be substantially improved by providing a deeper phenotypic characterization of spheroids model (with or without miRs) as suggested by reviewers. Further experimentation is also required to better characterize and validate the conclusions draw by the manipulation of miR-424 and 29 -signaling pathways in spheroids.

Essential revisions:

1) It is unclear if the 3D spheroid assay is a suitable model for probing coordination of tip cell proliferation and migration in angiogenesis. The switch from quiescent to activated angiogenic endothelial cell phenotype is classically associated with (and dependent on) a global increase in proliferation and high levels of P-ERK in tip cells – both of which are not observed in the spheroid assay. Indeed, recent work elegantly demonstrates that mid to low levels of P-ERK drive proliferation in the mouse retinal vessels, whereas high levels reduce tip cell proliferation via induction of p21 (Pontes-Quero, Nat. Comm. 2019). However, Rosano et al. indicate that P-ERK is repressed and dispensable in spheroid angiogenesis. Although the authors point out key differences between their 2D and 3D assays, such discrepancies between the spheroid assay and in vivo observations are overlooked. In particular:

- Why is ERK dispensable in this system but indispensable in vivo?

- Why are such low levels of proliferation (<4% of cells versus >90% in vivo) observed and why so variable between experiments (4% in Figure 2F versus 0.42% in Figure 6E – rising to only 2% upon DICER KD – i.e. below control levels of Figure 2F)? Do both tip and stalk cells not proliferate in this system? Considering spheroids are treated with VEGF for 18 hours, could the short 2 hour EdU pulse simply miss an earlier synchronized burst of VEGF-induced mitotic events?

- What is the organization of tip and stalk cells and localization of tip markers in this assay? Would a lack of stalk signatures upon VEGF stimulation suggest a disorganized structure? Does manipulation of Notch-mediated tip versus stalk identity generate predictable shifts in proliferation, P-ERK and miRNA expression? This information is key to confirming that VEGF-regulated miRNAs drive a tip (not stalk) phenotype, as proposed.

- Are phenotypes observed in the spheroid assay dependent or independent of p21?

2) How were the p38 and ERK selected? Evidently, the logical thing would have been to select those based on targets of those pathways being enriched in the Figure 3D but this is unclear from the manuscript.

3) Knockdown of DICER evidently has dramatic genome-wide effects through various pathways so it is impossible to rule out effects that act through other pathways than ERK. This should be clarified.

4) Can the authors prove that proliferation is promoted in the Dicer KO 3D cells using PCNA or cell cycle analysis via flow cytometer? It would be helpful to have some experiments that analyze the phenotype of the spheroid rather than the gene expression only.

5) Loss of Dicer promote the level of proliferative genes and inhibit SA, suggesting that the activation of proliferation inhibits SA – that is why it needs to be actively inhibited. However, I don't understand why inhibiting proliferation (with ERK inhibitor) in Dicer KO spheroid further reduces SA. The prediction is that SA in Dicer Ko should be rescued by the inhibition of proliferation, right? Why do they see a further decrease in absence of miRNAs?

6) I found the differences between Dicer/VEGF interaction in 3D vs. 2D model very interesting. Can the authors test if components of the miRNA machinery are differentially regulated by VEGF-A treatment in 3D vs. 2D? This should be easily extrapolated by their RNA seq data, or it could be easily done by qPCR with a time course of treatment with VEGF-A. I think that it would be interesting to know if vega regulates miRNAs biogenesis to mediate such a switch, or it is an innate response from the 3D vs. 2D cell behaviors? Perhaps some comments on this in the Discussion would be useful.

7) In general, the interpretation of proliferation vs. migration signaling in SA and why these should be regulated in 3D vs. 2D it is not clear. Do the authors believe that proliferation and migration need to be buffered to promote SA? Do they think that they need to be differentially regulated in time? For example, do they think that miRNA first stop proliferation then miRNA release migration? A clearer model that supports their main conclusion would be critical to follow the interpretation of these results.

8) The role of miR-424 and miR-29a have already been established in angiogenesis so the real novelty lies in identifying the target gene networks through which the miRNAs act. However, the validations are incomplete as only experimental evidence of miRNA mimics affecting gene expression (Figure 7D, E) is shown and the inhibitor data is missing. This missing experimental evidence should be provided.

9) Currently the experimental evidence provided under section "Biological validation of the miRNA Hub network in regulating Sprouting Angiogenesis" is insufficient to support the claim. To be able to present in an unbiased way that the miRNAs are regulating the computationally predicted networks, the authors need to perform RNA-seq analysis of mRNAs upon miRNA perturbation. This would significantly strengthen the manuscript that currently lies on predictions for miRNA effects and insufficient validation of few selected candidates.

10) miRNA-mRNA pairwise correlation demonstrated that miR-424-5p expression is induced and the targets (MAPK and cell cycle related) are repressed whereas the miR-29a-3p was downregulated and target mRNAs (ECM-related) upregulated. The angiogenesis assay (Figure 7D), however, does not support this. This puts in question the 3D model and whether it is more prone to any perturbation. Would you see an opposite effect in proliferation assay or in 2D angiogenesis model?

11) The functions of identified miRNA networks in angiogenesis are mostly implied rather than directly tested. As such, additional experimentation providing mechanistic insight would greatly improve the manuscript. For example:

- What is the function of modulating MAPK genes identified in Figure 5? Are P-ERK levels perturbed in VEGF-treated spheroids, as predicted?

- Which miRNAs are responsible for regulation of ERK pathway genes in Figure 5 and can modulation of these miRNAs predicable impact sprouting, P-ERK levels and proliferation?

- Likewise, does DICER impact P-ERK levels, as predicted in Figure 6?

- In Figure 7, does miR-inhibition increase target expression?

- Does miR mimic/inhibitor expression predictably increase/decrease proliferation and P-ERK?

- If so, can ERK inhibition further modulate effects of miR 424-5p? But not 29a-3p?

- It is stated that these miRNAs allow proliferation and migration to be "spatially segregated in different EC populations" but how would this be achieved? Are they expressed in distinct tip or stalk domains?

- If cells expressing miR 424-5p/29a-3p mimic/inhibitor are mixed with WT cells, how well do they compete for the tip position?

12) The relevance of last Results paragraph? I'd suggest the authors revise the text to better convey the main message. For example, "Furthermore, average expression of genes belonging to the up-regulated module identified in the network analysis showed a significant positive correlation with endothelial score." What does this mean? Please explain the endothelial score in the text. What is the relevance of findings i.e. is the effect specific to CRC or would some cancers not show enrichment for angiogenesis related gene modules?

13) The Discussion is confusing and would profit from more structure (separate paragraphs etc).

[Editors' note: further revisions were suggested prior to acceptance, as described below.]

Thank you for resubmitting your work entitled "A regulatory microRNA network controls endothelial cell phenotypic switch during sprouting angiogenesis" for further consideration by *eLife*. Your revised article has been evaluated by Didier Stainier as the Senior Editor, a Reviewing Editor and two reviewers.

The authors have improved the manuscript by clarifying many questions from the reviewers' comments and providing extensive further experimental evidence to support the conclusions. There are some points, however, that still need to be addressed.

1) To provide further validation of the miRNA-regulated networks (reply to reviewer comments 8-9), the authors demonstrated that up- or down-modulation of miR-424 and miR-29a significantly regulated several other miRNAs. The data, however, is not incorporated in the manuscript. Please include the data in the manuscript, to strengthen the conclusion of these two miRNAs representing central players of the transcriptional response.

2) Similarly, please provide the results evaluating the miR-424-5p and MiR-29-3p target expression upon inhibitor treatment (reply to reviewer comment 11).

3) The proliferation levels of < 0.5% makes it impossible to evaluate if there are any differences in SPHC and SPHV or to demonstrate any effect of the manipulation of miR-424-5p and MiR-29-3p levels. Therefore, the authors should avoid to overly interpret the data and discuss the limitations. The authors should also provide the information about the miR-424-5p and MiR-29-3p mimics/inhibitors not affecting proliferation in the 3D model to complement the results shown for the 2D model (Figure 7—figure supplement 3) that currently was only mentioned in the response letter.

4) Figure 4. Please explain which is the upregulated and downregulated hub in the figure legend.

5) VEGF does further repress pro-proliferative genes, but this is not a phenotypic switch as there is no change in proliferative phenotype. Also, the function of this repression of pro-proliferative genes is unknown, as 424-5p miR inhibitor fails to increase proliferation. The subtle increase in EC proliferation (and pro-proliferative genes) seen upon DICER KD is not evidence that miRs block proliferation upon VEGF treatment, as only VEGF-treated cells are shown. Are these also elevated in non-VEGF-treated cells? The authors should show the cytofluorimetric analysis of EdU incorporation into the DNA in DICER-WT and DICER-KD spheroids without VEGF-A (Figure 6E) to support that the effects of Dicer are VEGF-A depended.

6) In addition, the dispensability of proliferation and ERK in spheroid SA does not demonstrate that they essentially have to be repressed to enable SA in WT cells. Overall, there is insufficient evidence to support profound statements such as "the first SA step requires a miRNA-mediated inhibition of cell proliferation", or that migration and proliferation pathways "cannot be activated at the same time". Please revise the statement.

---

## [Author Response]

The reviewers raise a number of concerns that must be addressed before the paper can be accepted. There is some skepticism on how the data discovered with the spheroids 3D vs. 2D models can be relevant for the scientific community that study sprouting angiogenesis in vivo. This point could be substantially improved by providing a deeper phenotypic characterization of spheroids model (with or without miRs) as suggested by reviewers. Further experimentation is also required to better characterize and validate the conclusions draw by the manipulation of miR-424 and 29 -signaling pathways in spheroids.Essential revisions:1) It is unclear if the 3D spheroid assay is a suitable model for probing coordination of tip cell proliferation and migration in angiogenesis. The switch from quiescent to activated angiogenic endothelial cell phenotype is classically associated with (and dependent on) a global increase in proliferation and high levels of P-ERK in tip cells – both of which are not observed in the spheroid assay. Indeed, recent work elegantly demonstrates that mid to low levels of P-ERK drive proliferation in the mouse retinal vessels, whereas high levels reduce tip cell proliferation via induction of p21 (Pontes-Quero, Nat. Comm. 2019). However, Rosano et al. indicate that P-ERK is repressed and dispensable in spheroid angiogenesis. Although the authors point out key differences between their 2D and 3D assays, such discrepancies between the spheroid assay and in vivo observations are overlooked. In particular:- Why is ERK dispensable in this system but indispensable in vivo?

Spheroid assay exploited in this manuscript is based on the capability of endothelial cells to sense an angiogenic stimulus in a 3D gel of collagen I. According to several published observations, this assay supports the formation of tip cells that precedes the elongation of individual capillary-like sprouts originating from single spheroids (Heiss et al., 2015). Indeed, this assay has been used to demonstrate the inhibitory effects of molecules on tip cell formation (Adam et al., 2013). The gene expression heatmap presented in Figure 1B confirms at molecular level these observations, showing that VEGF-stimulated spheroids are enriched in tip marker genes. Therefore, we are aware that this assay does not recapitulate the complexity of the tip-stalk dynamics occurring in in vivo sprouting angiogenesis; nonetheless, it is a powerful tool to understand the effect of angiogenic inducers in the first events of capillary nascence, which are characterized by the lateral inhibition-driven tip cell selection (Eilken et al., 2010). The limit of our assay is now specified in the last paragraph of the Introduction.

Regarding the role of ERK in VEGF signaling, the first version of the manuscript reported that treatment with the ERK inhibitor SCH 772984 did not alter sprout activity, suggesting a negligible involvement of ERK in sprouting. We also reported that spheroid stimulation by VEGF resulted in reduced expression of some members of MAPK family, but we did not evaluate the level of P-ERK. Following the reviewer’s request, the current version of the manuscript now reports the P-ERK amount in control spheroids and in VEGF-stimulated spheroids, evaluated by Mesoscale technology (see Materials and methods section). We demonstrated that VEGF stimulation decreases the basal level of P-ERK (Figure 5C). To better elucidate the correlation between ERK function and the predicted role of miR-mediated post-transcriptional control of SA, in the first version of the manuscript we analyzed MAPK1 expression in DICER-silenced endothelial cells and showed that its expression increased (Figure 6A). In the present version of the manuscript, we also analyzed P-ERK levels in DICER-silenced endothelial cells and showed that DICER silencing resulted in an increased amount of P-ERK (Figure 6B), thus supporting the observed sensitivity of the assay to ERK inhibitor as reported in the first version of the manuscript (first version Figure 6B, C, now Figure 6C, D). Furthermore, differential expression analysis performed with DESeq2 showed that CDKN1A (p21) mRNA is not differentially expressed (log_2_(Fold Change) = -0.17, P value adjusted = 0.68), and p21 inhibition by the selective drug UC2288 did not have any effect on sprouting (Author response image 1). This indicates that sprouting, in this model and at the stage considered, does not depend on p21 activity.

We agree that our data do not support the recent observations published in Nature Communications by Benedicto’s group, which performed in vivo retina studies in mice. This discrepancy is now discussed in the Discussion section according to these points: (i) the different time scale characterizing the two models (hours versus days); (ii) the feature of our in vitro model, which precisely describes the early tip selection from a cell monolayer and its motion and cannot take into account other co-regulatory events characterizing SA, in comparison to the retina model where neurons play a relevant role in establishing available amount of VEGF (Okabe et al., 2014); (iii) the inability of the spheroid model to analyze tip-stalk dynamics.

- Why are such low levels of proliferation (<4% of cells versus >90% in vivo) observed and why so variable between experiments (4% in Figure 2F versus 0.42% in Figure 6E – rising to only 2% upon DICER KD – i.e. below control levels of Figure 2F)?

We apologize for a mistake that happened during the uploading of the figures. The current manuscript contains the correct figure that shows a 0.04% of proliferating cells in SPHC and a 0.12% of proliferating cells in SPHV. (The correct figure was indeed already included in the initial submission version of the manuscript).

Do both tip and stalk cells not proliferate in this system?

As explained above, the SA model utilized here is not suitable to analyze stalk cell behavior, nor to study the interplay between tip and stalk cells. This limit is most probably due to the short life time of endothelial cells in these experimental conditions. This assumption is supported by the following observations:

i) RNAseq analysis clearly demonstrates that stalk specific genes are down-modulated in our experimental conditions (Figure 1B);

ii) DLL4 (tip cell marker) immunofluorescence shows a strong labeling of all endothelial cell sprouts (Figure 1—figure supplement 2A);

iii) Spheroids generated with endothelial cells transduced with a fluorescent reporter of Notch activity showed that very few cells activated the Notch pathway. These cells mainly localized within the spheroid core, therefore cannot be considered stalk cells. An image representative of this experiment has been included in the current version of the manuscript in Figure 1—figure supplement 2A).

Having showed that the spheroid model generates mainly tip cells and that the number of stalk cells is negligible (see below), and because of the short period of time took into account (18 h) compared to the retina model, we believe that substantial cell proliferation in this initial phase of SA is not to be expected.

Considering spheroids are treated with VEGF for 18 hours, could the short 2 hour EdU pulse simply miss an earlier synchronized burst of VEGF-induced mitotic events?

To exclude that low proliferation rate observed after VEGF stimulation was simply depending on the temporal window chosen for the EdU pulse, we analyzed cell proliferation by pulsing cells at 2, 6 or 12 hours. As shown Author response image 2, cell proliferation remained under 1% in all the 4 experimental conditions analyzed, indicating that the reported data in Figure 2F are not an underestimation due to an incorrect choice of the time-point took into account.

**Author response image 2. respfig2:** 

- What is the organization of tip and stalk cells and localization of tip markers in this assay?

Besides the gene expression data above mentioned (Figure 1B), we studied the localization of DLL4 and CXCR4, and by using a fluorescent CBF1/RBP-J reporter we evaluated Notch pathway activity, which respectively are considered markers of tip and stalk cells (Moya et al., 2012). VEGF stimulated spheroids were fixed, permeabilized and stained with Ab anti-DLL4, or the DsRed fluorescent protein, indicative of Notch activity, was visualized. Figure 1—figure supplement 2A, which is representative of 16 different spheroids from two independent experiments, clearly shows that DLL4 and CXCR4 staining are predominant, while very few cells with active Notch signaling pathway (white arrow) contributed to the formation of endothelial sprouts, indicating that the model is markedly oriented toward the tip phenotype. CXCR4 immunofluorescence and Notch reporter experiment have been added to Figure 1—figure supplement 2A.

Would a lack of stalk signatures upon VEGF stimulation suggest a disorganized structure? Does manipulation of Notch-mediated tip versus stalk identity generate predictable shifts in proliferation, P-ERK and miRNA expression? This information is key to confirming that VEGF-regulated miRNAs drive a tip (not stalk) phenotype, as proposed.

Manipulation of Notch pathway was achieved by treating spheroids with the γ-secretase inhibitor DAPT or with the Notch ligand DLL4, which respectively halts and triggers this pathway (Jakobsson et al., 2010; Shah et al., 2017). Spheroids obtained under these two conditions were subjected to i) morphometric analysis, ii) miR-424-5p and miR-29a-3p expression analysis, iii) P-ERK measurement, iv) proliferation assay.

The results obtained are summarized in Author response table 1:

TreatmentsproutingmiR-424miR-29aP-ERKProliferationVEGF+DAPTincreasedincreasedreduceddecreasedNot affectedVEGF+DLL4reducedreducedincreasedunaffectedNot affected

i) Effects of Notch pathway manipulation on spheroids morphology and corresponding quantification of sprouting are now reported in Figure 1—figure supplement 2B. Notch pathway inhibition by DAPT, which favors the tip phenotype (Jakobsson et al., 2010), increased sprouting, while the opposite was obtained by treatment with exogenous DLL4.

ii) DAPT treatment resulted in increased expression of miR-424-5p and decreased expression of miR-29a-3p. The opposite effect was obtained with DLL4 treatment. This further supports the concept that miR-424-5p and miR-29a-3p expression is regulated during tip cells specification. In fact, in our model miR-424-5p expression increases while miR-29a-3p expression decreases. Data about miR-424-5p and miR-29a-3p expression after Notch pathway manipulation are now reported in Figure 7C, D.

iii) We analyzed the effect of Notch pathway manipulation on of P-ERK levels in SPHV, measured by Meso Scale Discovery (MSD) technology and expressed as ratio of P-ERK/total ERK. The data obtained are shown in Author response image 3. According with our envisaged scenario where inhibition of ERK pathway by miRNAs occurs during tip cells selection, Notch pathway inhibition by DAPT, which increases tip cells, further decreases P-ERK in SPHV. However, DLL4 stimulation does not result in a mirrored effect as reported for tip selection and miRNAs modulation.

**Author response image 3. respfig3:** 

We underline that the complexity of signaling pathways, their reciprocal interferences, and the methods here used to modulate Notch pathway (a ligand versus a chemical inhibitor) can explain the incoherence between these results. Furthermore, the analysis of single signaling event (i.e. P-ERK) does not necessarily match an observed biological function (i.e. sprouting) because of the well-established occurrence of crosstalks, referring to the case where two inputs work through distinct signaling pathways but cooperate to regulate the cellular work. Therefore, we decided to not show some of these results because, in our opinion, they do not modify the general meaning of the other data obtained by Notch modulation, which confirm that VEGF triggers a tip phenotype in the early phase of SA by a miR-dependent mechanism. To explain the observed discrepancy between DDL4 and DAPT effect on VEGF mediated P-ERK level in tip cells, more experiments would be required, which are out of the scope of this manuscript.

iv) We analyzed cell proliferation levels by flow cytometry in SPHV treated with DAPT or DLL4. Cell proliferation was not significantly affected (Author response image 4). Considering the data shown in Figure 1 and Figure 1—figure supplement 2, we can explain these results as follows:

i) Stalk cells population, which should represent the proliferating component, in this model is negligible. Therefore, treatment with DAPT favors tip cell selection more than could actually decrease stalk cells number. In this situation, it is not possible to expect a reduction in cell proliferation rate which is already close to zero.

ii) DLL4 treatment reduced sprouting, meaning a reduction in tip cells rather than an increase in stalk cells number, which can explain the lack of a proliferative response.

**Author response image 4. respfig4:** 

- Are phenotypes observed in the spheroid assay dependent or independent of p21?

We performed the sprouting assay in the presence of increasing concentrations of the p21-specific chemical inhibitor UC2288. Quantifications of the sprout area did not indicate any effect on sprouting when p21 is suppressed, as reported in Author response image 1 (average of two experiments ± SEM).

2) How were the p38 and ERK selected? Evidently, the logical thing would have been to select those based on targets of those pathways being enriched in the Figure 3D but this is unclear from the manuscript.

The main observations taken into account were the strong down-regulation of genes controlling cell proliferation and the fact that MAPK1 (an upstream regulator of cell proliferation genes) is one of the genes most targeted by miRNAs in the whole network. The GO analysis presented in Figure 3D showed a selection of few (of many) GO terms with the lowest P-values. Indeed, the “MAPK cascade” term was present in the output of this analysis, and associated with the “blue” cluster. This information has been added to Figure 3D. MAPK1 and other MAP kinases constitute the “MAPK cascade” GO term, while p38 upstream activators are shared with the “*JNK* cascade”, which is associated with the “green” cluster in Figure 3D. Furthermore, p38 and ERK were selected on the basis of a huge number of papers that demonstrate that p38 and ERK represent two intermediate nodes of VEGF receptor signaling pathway able to switch the input respectively towards a motogenic or mitogenic activity. Excellent reviews overview this concept (Koch et al., 2011; Simons et al., 2016). The whole paragraph and Figure 3 have been reshaped for improved clarity.

3) Knockdown of DICER evidently has dramatic genome-wide effects through various pathways so it is impossible to rule out effects that act through other pathways than ERK. This should be clarified.

The experiments based on DICER silencing were planned to support the concept that miR-mediated post-transcriptional control is instrumental to SA and they have to be considered as complementary to the GOF and LOF experiments specifically targeting miR-29a-3p and miR-424-5p. The concept that DICER inhibition can affect multiple pathways is now clarified in the Discussion section at the end of the paragraph ‘Regulation of MAP Kinase activity by miRNAs’.

4) Can the authors prove that proliferation is promoted in the Dicer KO 3D cells using PCNA or cell cycle analysis via flow cytometer? It would be helpful to have some experiments that analyze the phenotype of the spheroid rather than the gene expression only.

We respectfully underline that in the first version of the manuscript we measured the percentage of proliferating cells by EdU detection and flow cytometry (Figure 6E of the first version, now Figure 6F). In this experiment, as well as in all other experiments where we measured cell proliferation, the BrdU analog EdU is added to the cell culture where it is incorporated into the newly synthetized DNA in cells that are actively proliferating. The percentage of EdU positive cells therefore corresponds to the percentage of cells being in the S phase of the cell cycle.

5) Loss of Dicer promote the level of proliferative genes and inhibit SA, suggesting that the activation of proliferation inhibits SA – that is why it needs to be actively inhibited. However, I don't understand why inhibiting proliferation (with ERK inhibitor) in Dicer KO spheroid further reduces SA. The prediction is that SA in Dicer KO should be rescued by the inhibition of proliferation, right? Why do they see a further decrease in absence of miRNAs?

We showed that loss of DICER increased cells proliferation. However, the lack of miRNA had an extensive effect on the transcriptome, due to their broad activity. For instance, DICER silencing caused a general downregulation of cell adhesion molecules (Author response image 5, from the GSEA analysis of SPHV_DICER^KD^ versus SPHV_DICER^WT^), which can account for reduced cell motility. We currently cannot rule out that the extensive rewiring of the network that sustains sprouting caused by the lack of miRNAs could benefit, to a certain extent, from increased ERK activation, and therefore making the system sensitive to ERK inhibition, i.e. presence of proliferating cells in the sprouts.

A second point to be considered is related to methodological aspects of our assay characterized by the presence of homotypic cellular junctions, which mediate a contact inhibition of cell proliferation. The stimulation of spheroids with VEGF triggers cell motility, loss of endothelial-endothelial junctions and in principle it should allow cell proliferation. The findings of this paper (e.g. miR-mediated arrest of cell-cycle) predict that the sequence migration/proliferation in sprouting angiogenesis is temporally regulated and reflects the tip/stalk dynamics occurring in vivo.

However, in this in vitro assay, the block of miRNA machinery by DICER silencing inhibits the motility phase of sprouting angiogenesis and subsequently impairs the cell-cell junction dismantling. This condition may mask the theoretical expected result of ERK inhibition in DICER-silenced cells.

**Author response image 5. respfig5:** 

6) I found the differences between Dicer/VEGF interaction in 3D vs. 2D model very interesting. Can the authors test if components of the miRNA machinery are differentially regulated by VEGF-A treatment in 3D vs. 2D? This should be easily extrapolated by their RNA seq data, or it could be easily done by qPCR with a time course of treatment with VEGF-A. I think that it would be interesting to know if vega regulates miRNAs biogenesis to mediate such a switch, or it is an innate response from the 3D vs. 2D cell behaviors? Perhaps some comments on this in the Discussion would be useful.

From the expression datasets from 2D or 3D cells stimulated with VEGF we extracted information about genes involved in miRNA biogenesis (DGCR8, DICER1, DROSHA, TARBP2, XPO5) or in miRNA-target recognition (AGO1-4). The results of the differential expression (DE) analysis performed with LIMMA (Author response image 6) do not indicate any statistically significant change in these gene’s expression, either in 2D or 3D condition (fold change >0.5 and adjusted P value <0.05 for a gene to be considered differentially expressed).

**Author response image 6. respfig6:** 

This would suggest that VEGF regulates miRNAs expression at the transcriptional level, although we cannot rule out the existence of post-transcriptional modifications (i.e. phosphorylations) occurring to the proteins that constitute the miRNA machinery (Treiber et al., 2019), which could alter the catalytic activity of some of its components. It would be very interesting, therefore, to compare alterations of global miRNAs expression upon VEGF stimulation in 3D vs. 2D condition but we believe it is beyond the main scope of this paper. We did, nonetheless, measure the expression of miR-424-5p and miR-29a-3p by real-time PCR in HUVECs cultured in 2D (H2D) and stimulated with VEGF (Author response image 7, average of n=3 experiments ± SEM). In this condition, where VEGF triggers cell proliferation, miR-424-5p expression decreased after VEGF stimulation, as opposite to the spheroid model. miR-29a-3p expression was downregulated as in spheroids, in agreement with the fact that VEGF stimulation in 2D culture also activates cell scatter migration.

**Author response image 7. respfig7:** 

7) In general, the interpretation of proliferation vs. migration signaling in SA and why these should be regulated in 3D vs. 2D it is not clear. Do the authors believe that proliferation and migration need to be buffered to promote SA? Do they think that they need to be differentially regulated in time? For example, do they think that miRNA first stop proliferation then miRNA release migration? A clearer model that supports their main conclusion would be critical to follow the interpretation of these results.

2D culture models cannot recapitulate the complexity of the in vivo environment. In a 2D culture of ECs stimulated with VEGF, proliferating cells coexist with cells with increased motility, due to dual role of VEGF on ECs as activator of both cell proliferation and cell migration. In the 3D spheroid model, however, being closer to an in vivo condition, it was possible to observe that cells that are initially contact-inhibited are induced to migrate by VEGF, while not entering in a proliferative state. We believe that miRNAs, by repressing proliferation genes, act in this phase as a switch that allows endothelial cells to enter the tip phenotype. A different (stalk) phenotype characterized by cell proliferation would emerge at a later point. It would be very interesting, therefore, to analyze miRNAs activity in stalk cells, but unfortunately in this model they are not enough represented.

8) The role of miR-424 and miR-29a have already been established in angiogenesis so the real novelty lies in identifying the target gene networks through which the miRNAs act. However, the validations are incomplete as only experimental evidence of miRNA mimics affecting gene expression (Figure 7D, E) is shown and the inhibitor data is missing. This missing experimental evidence should be provided.9) Currently the experimental evidence provided under section "Biological validation of the miRNA Hub network in regulating Sprouting Angiogenesis" is insufficient to support the claim. To be able to present in an unbiased way that the miRNAs are regulating the computationally predicted networks, the authors need to perform RNA-seq analysis of mRNAs upon miRNA perturbation. This would significantly strengthen the manuscript that currently lies on predictions for miRNA effects and insufficient validation of few selected candidates.

Points 8 and 9 raised by the reviewers concern very important aspects of the biology and functioning of miRNA-mediated regulatory networks. We decided to address point 8 and 9 together because, in our opinion, the requested experiments need to be discussed taking into account the general rules by which miRNA-mediated mechanisms occur in living cells and are common to the two points. Actually, it is well demonstrated that the principles of degeneracy and pluripotentiality of miRNA function imply that individual protein coding genes can be targeted by several miRNAs whereas a single miRNA may concomitantly regulate a subset of several different protein coding genes. We believe that the results presented in our work have to be interpreted in agreement to this principle. Before analyzing by mRNA-seq the SA gene network in endothelial cells in which miR-424 and 29a would be perturbed, to answer to point 9 issue we decided preliminarily to investigate the impact of GOF and LOF approaches targeting miR-424 and 29a on miRNA expression. To this purpose, by means of miRNA-PCR cards we studied the global effect of miRNA perturbation on the expression levels of a large panel of miRNAs, selected among those that are endogenously expressed in our model. The entire procedure was performed for the GOF of miR-424 and miR-29a, as well as for the LOF of the same two miRNAs. Strikingly, results point out a vast reshaping of endogenous miRNA levels upon the modulation of the two “hub” miRNAs, miR-424 and 29a, thus indicating that miR-424 and miR-29a modulation induces a deep re-wiring of the global miRNA network in addition to a modulation of their target coding genes. These miRNA-miRNA modulations are probably due to indirect interactions, e.g. TF-mediated. As example of the results, we show in Author response image 8 heatmap reporting the top-differentially expressed miRNAs (| logFC | > 2.0) upon up- or down-modulation of miR-424 and miR-29a, having restricted the list to the miRNAs present in the two “hub” modules – in other words, modulated miRNAs that share the same targets with miR-424-5p or miR-29a-3p.

**Author response image 8. respfig8:** 

The above experiments also highlighted the opposite effect on the modulated miRNAs between the GOF or LOF experiments, thus indicating a non-specular trend following the up-modulation or down-modulation procedure. The global reshaping of the miRNAs observed is likely to propagate to the downstream regulated target coding genes.

Overall, the interpretation of these results prompted us to stress a high interconnection among miRNA themselves and thus the miRNA-gene network, making it arduous to isolate the effect of a single miRNA from the global behavior of the whole miRNA network by means of a high-throughput mRNA-seq, at least in our system. On the basis of these results the mRNA-seq experiment suggested at point 9 seems not so robust to validate the predicted network, since our miRNA-card experiments demonstrate that the perturbation of one miRNA can reset the whole miRNA network and indeed change in unpredictable ways the network of differential expressed coding genes. In other words, the expected results of a mRNA-seq experiment upon GOF or LOF of miR-424 and miR-29a would probably require fully recapitulating the behavior of their target genes, and the discussion of such indirect interactions. We believe this is a very interesting observation that will need specific effort to be explored but it concerns the biology of miRNA rather than the control of SA and we plan to address this issue in future studies. Inside this scenario, we consider a major point in our work the results reported in the section “DICER Knock-Down Rescues VEGF-A Proliferative Effect”, where we used a DICER Knock-Down strategy to illustrate the rescue of the sprouting phenotype rather than the effect of a single miRNA. At the same time, the biological validations reported in the section “Biological validation of the miRNA Hub network regulating Sprouting Angiogenesis” illustrate the role of miR-424 and miR-29a as overall capable of sustain the observed phenotype upon up- or down-modulation. Of note, by real-time PCR, expression of 10 predicted target genes for miR-424-5p and 10 for miR-29a-3p were already validated in the current version of our manuscript. To partially overcome this issue, thinking also to reviewer’s point 8, we investigated publicly available databases (namely MirTarbase, DIANA Tarbase and StarBase) of experimentally validated miRNA-gene interactions for the presence of miR-424 and miR-29a target genes among the ones present in our two hubs. The results are the following:

For miR-424:

97 out of 98 of the miR-424 hub target genes are validated in the above databases (24 of them validated by all the three databases, whilst 73 of them are validated by at least one).

For miR-29a:

23 out of 25 of the miR-29a target genes are validated in the above databases (7 of them validated by all the three databases, whilst 16 of them are validated by at least one).

This information has been added in the Results section ‘Biological validation of the miRNA Hub Network Regulating Sprouting Angiogenesis’ and as Table 1—source data 1.

Summing up the above analysis, we can say that > 90% of the miR-424 and miR-29a target genes present in our two hubs are already experimentally validated in some model systems, hence supporting our answer to the reviewer’s question regarding insufficient validation of selected targets. We believe that the lack of reciprocity between miRNA inhibitors and mimics in regulating coding genes depends on the miRNA features above summarized. Therefore, we propose not to show these data in the main text, planning to discuss these issued in a separate paper. We are thankful again for the reviewer’s comments. To address these concerns, the main text was updated in the in the Results section and in the Discussion section basically avoiding the usage of terms indicating the uniqueness of direct interactions among the miRNA-network we presented.

10) miRNA-mRNA pairwise correlation demonstrated that miR-424-5p expression is induced and the targets (MAPK and cell cycle related) are repressed whereas the miR-29a-3p was downregulated and target mRNAs (ECM-related) upregulated. The angiogenesis assay (Figure 7D), however, does not support this. This puts in question the 3D model and whether it is more prone to any perturbation.

To exclude unspecific effects induced by the GOF/LOF strategy targeting miR-424-5p and miR-29a-3p, which could be accountable for a general sensitivity of the system to any perturbation, we used the same strategy to modulate miR-16-5p expression. The results are shown in Figure 7—figure supplement 1F, G, and indicate that miR-16-5p GOF/LOF did not have any relevant effect on spheroids sprouting. We chose specifically miR-16-5p because, other than having a recognized role in angiogenesis, belongs to the same family miR-424-5p belongs, having in common the same seed region. This implies that miR-424-5p and miR-16-5p can, in principle, target the same set of genes, according to TargetScan predictions. However, our bioinformatics analysis, which integrates TargetScan predictions with expression data, predicted a different subset of target genes for miR-424-5p and miR-16-5p in our 3D model (Figure 4—source data 1). We believe that this result is consistent with the idea that miRNA activity on their predicted targets depends also on the global cellular context, i.e. expression levels of miRNAs and targets availability. Furthermore, miR-424-5p and miR-29a-3p, having a high number of connections in our network, can be considered “hub” miRNAs, while miR-16-5p is a more peripheral miRNA. According to the network theory, disturbance of a hub, not of a peripheral node, is expected to cause a dramatic effect on the whole system, and that is actually what we observed. This experiment therefore demonstrates that the effects of miR-424-5p and miR-29a-3p GOF/LOF are specific, and reinforces the effectiveness of our bioinformatics analyses. This information has now been clarified in the manuscript.

Would you see an opposite effect in proliferation assay or in 2D angiogenesis model?

We performed a proliferation assay using HUVECs treated with miRNA mimics or inhibitors, cultured in 2D and stimulated with VEGF for the same time as spheroids (18h). We observed a reduction of proliferation in cells treated with miR-424-5p mimic, while miR-424-5p inhibitor was not significantly effective. miR-29a-3p mimic and inhibitor were both not effective.

We also performed a migration assay by using the xCELLigence real time cell analyzer, in which VEGF was used as chemoattractant for HUVECs treated with miRNA mimics or inhibitors. In this experiment, miR-424-5p mimic had a positive effect on cell migration, while miR-29a-3p mimic had a negative effect, in agreement with its predicted role of negative regulator of cell migration. However, we did not obtain any effect in cells treated with miRNA inhibitors.

These data have been added to the manuscript as Figure 7—figure supplement 3.

11) The functions of identified miRNA networks in angiogenesis are mostly implied rather than directly tested. As such, additional experimentation providing mechanistic insight would greatly improve the manuscript. For example:- What is the function of modulating MAPK genes identified in Figure 5?

According to what explained in point 2, we decided to modulate the activity of ERK and p38 because they constitute the two nodes of VEGF pathway respectively involved in directing cell work towards proliferation or migration. In fact, the ERK inhibitor targets the protein product of MAPK1 gene ERK2, while the p38 inhibitor targets the protein product of MAPK14 gene p38α.

Are P-ERK levels perturbed in VEGF-treated spheroids, as predicted?

A new experiment is shown in Figure 5C demonstrating that VEGF reduces P-ERK levels in our experimental model.

- Which miRNAs are responsible for regulation of ERK pathway genes in Figure 5 and can modulation of these miRNAs predicable impact sprouting, P-ERK levels and proliferation?

In principle, all the miRNAs represented in Figure 5 can contribute to the post-transcriptional regulation of their respective MAPK targets. The network proposed here, in fact, integrates target prediction with expression data that were specifically obtained from the spheroid model. Our main goal here was to draw attention to how miRNAs can impact the regulation of a complex biological process (proliferation versus migration), rather than analyzing every single miRNA-target interaction, which would require a specific experimental approach due to the combinatorial nature and cumulative effect which is intrinsic of the miRNA activity. Some of the interaction showed here have been, however, already demonstrated, as mentioned in the Discussion section, such as miR-494/RAF1 or miR-424/MAP2K1. We did, however, show that expression of several MAPK target genes increases when miRNAs expression is reduced by DICER knock down, and we investigated more in detail the role of miR-424-5p, which is the miRNA with the highest number of connections (5) in the MAPK sub-network. Figure 7E shows the impact of miR-424-5p modulation on sprouting, Figure 7G shows the effect of miR-424-5p mimic on the MAPK target genes MAP2K1 and MAP3K3, and Figure 7—figure supplement 3 shows the effect of miR-424-5p mimic and inhibitor on P-ERK levels.

- Likewise, does DICER impact P-ERK levels, as predicted in Figure 6?

A new experiment is shown in Figure 6B demonstrating that DICER silencing increases P-ERK levels in VEGF-treated spheroids.

- In Figure 7, does miR-inhibition increase target expression?

We evaluated the expression of miR-424-5p and miR-29a-3p targets that were previously analyzed in the mimic experiments in spheroids transfected with the miRNA inhibitors. The results are shown in Author response image 9 (average of n=2 experiments ± SEM). We could observe upregulation in only a few target genes, while for most of the genes analyzed the miRNA inhibitor did not elicit a significant effect.

**Author response image 9. respfig9:** 

We then analyzed the expression of other miR-424-5p and miR-29a-3p targets that are present in their respective miRNA subnetworks and that were not previously analyzed (Author response image 10, average of n=2 experiments ± SEM). For most of these target genes, we could observe a significant upregulation upon miRNAs inhibition.

**Author response image 10. respfig10:** 

As discussed in the reply to points 8 and 9, miRNAs modulation by miRNA mimics or inhibitors causes significant alterations to many other miRNAs’ expression levels. The heatmaps in Author response image 11 show more in detail that inhibition of miR-424-5p and mir-29a-3p significantly increases the expression of several miRNAs that are represented in the sprouting network and share the same targets with miR-424-5p or mir-29a-3p.

**Author response image 11. respfig11:** 

We believe this could explain why inhibition of a single miRNA does not result in a significant increase of expression of all its predicted targets in a predictable way, also considering that each single gene is targeted by several other miRNAs. This might also explain why inhibition of miR-424-5p alone was not sufficient to revert the proliferative phenotype, while removal of miRNAs by DICER knock-down was effective. Another consideration is that transfection of miRNA inhibitors, although successful, do not completely abrogate miRNAs expression, as shown in Figure 7—figure supplement 1E.

- Does miR mimic/inhibitor expression predictably increase/decrease proliferation and P-ERK?

We performed the sprouting assay with cells transfected with miR-424-5p and mir-29a-3p mimics or inhibitors, and analyzed P-ERK levels by MSD technology. Figure 7—figure supplement 1I shows that only miR-424-5p manipulation could perturb the system, resulting in reduced P-ERK activity with miR-424-5p mimic and increased P-ERK activity with miR-424-5p inhibitor. This is in agreement with the predictions showed in the MAPK network (Figure 5A), where miR-424-5p interacts with, and should down-regulate, several MAPK upstream of ERK, while miR-29a-3p is not part of the MAPK network.

We did not, however, observe any effect on cell proliferation when miR-424-5p or miR-29a-3p expression was manipulated in spheroids by miRNA mimics or inhibitors. This can be explained by the fact that changes in P-ERK activity were modest, and by the fact that many different miRNAs are involved in the MAPK network (Figure 5A). Actually, only miR-424-5p inhibitor would predictably increase cell proliferation, which cannot be lowered further by miR-424-5p mimic. We did, however observe increased proliferation in spheroids DICER KD (Figure 6) and reduced proliferation in 2D cells treated with the miR-424-5p mimic.

- If so, can ERK inhibition further modulate effects of miR 424-5p? But not 29a-3p?

In the sprouting assay, treatment with ERK inhibitor did not further modulate effects of miRNA mimics or inhibitors (Author response image 12, average of n=2 experiments ± SEM).

**Author response image 12. respfig12:** 

- It is stated that these miRNAs allow proliferation and migration to be "spatially segregated in different EC populations" but how would this be achieved? Are they expressed in distinct tip or stalk domains?

Our experiments indicate that in the initial phase of SA miRNAs have an inhibitory effect on proliferative signals and contribute to the differentiation of non-proliferating tip cells. The phrase has been modified to “these two fundamental VEGF-A downstream pathways cannot be activated at the same time in tip cells” in the Discussion section.

- If cells expressing miR 424-5p/29a-3p mimic/inhibitor are mixed with WT cells, how well do they compete for the tip position?

Spheroid assay was performed by mixing in equal parts HUVECs transfected with miRNA mimics or inhibitors, or relative controls, and HUVECs transduced with a lentiviral vector for the stable expression of the fluorescent protein DsRed. Fluorescent cells at the very tip of the sprouts were counted. Figure 7—figure supplement 2 shows that cells transfected with either miR-424-5p or miR-29a-3p mimics or inhibitor were impaired in reaching the tip position. This observation is consistent with the data showed in Figure 7E, F, where alteration of miR-424-5p and miR-29a-3p expression impairs sprouting.

12) The relevance of last Results paragraph? I'd suggest the authors revise the text to better convey the main message. For example, "Furthermore, average expression of genes belonging to the up-regulated module identified in the network analysis showed a significant positive correlation with endothelial score." what does this mean? Please explain the endothelial score in the text. What is the relevance of findings i.e. is the effect specific to CRC or would some cancers not show enrichment for angiogenesis related gene modules?

The last paragraph has been extensively revised and the endothelial score explained. These findings could be extended to any other cancer whose growth depends on angiogenesis, or any cancer type for which anti-angiogenic therapy has been proven beneficial. This information has been added to the Discussion section.

13) The Discussion is confusing and would profit from more structure (separate paragraphs etc).

The Discussion has been divided into paragraphs, each with a title.

[Editors' note: further revisions were suggested prior to acceptance, as described below.]

The authors have improved the manuscript by clarifying many questions from the reviewers' comments and providing extensive further experimental evidence to support the conclusions. There are some points, however, that still need to be addressed.1) To provide further validation of the miRNA-regulated networks (Reply to reviewer comments 8-9), the authors demonstrated that up- or down-modulation of miR-424 and miR-29a significantly regulated several other miRNAs. The data, however, is not incorporated in the manuscript. Please include the data in the manuscript, to strengthen the conclusion of these two miRNAs representing central players of the transcriptional response.

A new paragraph entitled “Modulation of miRNA hubs alters network architecture” and a figure (Figure 8) describing the effect of miR-424-5p and miR-29a-3p modulation on other miRNAs have been added to the Results section. To reinforce the concept that miR-424-5p and miR-29a-3p modulation has a widespread effect on other miRNAs’ expression, we included a heatmap (Figure 8A) that was not initially included in the previous reply to reviewers. This heatmap shows the effect of miR-424-5p and miR-29a-3p modulation on all the miRNAs expressed in our model. The second heatmap (Figure 8B) focuses only on the miRNAs comprised in the SA network. The results have also been discussed.

2) Similarly, please provide the results evaluating the miR-424-5p and MiR-29-3p target expression upon inhibitor treatment.

Expression of miR-424-5p and miR-29a-3p targets upon inhibitor treatment have been added in Figure 7—figure supplement 4.

3) The proliferation levels of < 0.5% makes it impossible to evaluate if there are any differences in SPHC and SPHV or to demonstrate any effect of the manipulation of miR-424-5p and MiR-29-3p levels. Therefore, the authors should avoid to overly interpret the data and discuss the limitations. The authors should also provide the information about the miR-424-5p and MiR-29-3p mimics/inhibitors not affecting proliferation in the 3D model to complement the results shown for the 2D model (Figure 7—figure supplement 3) that currently was only mentioned in the response letter.

The information about the proliferation assay in the 3D model has been added as Figure 7—figure supplement 3A, and the limitations of the assay have been discussed. In the Discussion section, the phrases “Furthermore, miR-424-5p and miR-29a-3p target prediction can be an example of how the activity of a miRNA can be directed to a specific cellular function” and “extended the existing information about cell-autonomous function of these miRNAs thus identifying new molecular targets related to the endothelial role in cancer SA” have been removed.

4) Figure 4. Please explain which is the upregulated and downregulated hub in the figure legend.

The description of the two network components has been added to the figure legend.

5) VEGF does further repress pro-proliferative genes, but this is not a phenotypic switch as there is no change in proliferative phenotype. Also, the function of this repression of pro-proliferative genes is unknown, as 424-5p miR inhibitor fails to increase proliferation. The subtle increase in EC proliferation (and pro-proliferative genes) seen upon DICER KD is not evidence that miRs block proliferation upon VEGF treatment, as only VEGF-treated cells are shown. Are these also elevated in non-VEGF-treated cells? The authors should show the cytofluorimetric analysis of EdU incorporation into the DNA in DICER-WT and DICER-KD spheroids without VEGF-A (Figure 6E) to support that the effects of Dicer are VEGF-A depended.

GSEA data for the comparison of *DICER^KD^* versus *DICER^WT^* in non-stimulated spheroids has been added as Figure 6—figure supplement 1C (not significant). The cytofluorimetric analysis of spheroids not stimulated with VEGF-A (SPHC) has been added in Figure 6F, showing that *DICER^KD^* per se does not affect cell proliferation.

6) In addition, the dispensability of proliferation and ERK in spheroid SA does not demonstrate that they essentially have to be repressed to enable SA in WT cells. Overall, there is insufficient evidence to support profound statements such as "the first SA step requires a miRNA-mediated inhibition of cell proliferation", or that migration and proliferation pathways "cannot be activated at the same time". Please revise the statement.

The indicated statements have been revised.